# SDE2 integrates into the TIMELESS-TIPIN complex to protect stalled replication forks

Julie Rageul [1,6], Jennifer J. Park[1,6], Ping Ping Zeng[1], Eun-A Lee[2], Jihyeon Yang[2], Sunyoung Hwang[2], Natalie Lo[1], Alexandra S. Weinheimer[3], Orlando D. Schärer[2,4], Jung-Eun Yeo[2✉] & Hyungjin Kim [1,5✉]

Protecting replication fork integrity during DNA replication is essential for maintaining genome stability. Here, we report that SDE2, a PCNA-associated protein, plays a key role in maintaining active replication and counteracting replication stress by regulating the replication fork protection complex (FPC). SDE2 directly interacts with the FPC component TIMELESS (TIM) and enhances its stability, thereby aiding TIM localization to replication forks and the coordination of replisome progression. Like TIM deficiency, knockdown of SDE2 leads to impaired fork progression and stalled fork recovery, along with a failure to activate CHK1 phosphorylation. Moreover, loss of SDE2 or TIM results in an excessive MRE11-dependent degradation of reversed forks. Together, our study uncovers an essential role for SDE2 in maintaining genomic integrity by stabilizing the FPC and describes a new role for TIM in protecting stalled replication forks. We propose that TIM-mediated fork protection may represent a way to cooperate with BRCA-dependent fork stabilization.

[1] Department of Pharmacological Sciences, State University of New York at Stony Brook, Stony Brook, New York 11794, USA. [2] Center for Genomic Integrity, Institute for Basic Science, Ulsan 44919, Republic of Korea. [3] Department of Biochemistry and Cell Biology, State University of New York at Stony Brook, Stony Brook, New York 11794, USA. [4] Department of Biological Sciences, School of Life Sciences, Ulsan National Institute of Science and Technology, Ulsan 44919, Republic of Korea. [5] Stony Brook Cancer Center, Renaissance School of Medicine at Stony Brook University, Stony Brook, New York 11794, USA. [6] These authors contributed equally: Julie Rageul, Jennifer J. Park. ✉email: jyeo@ibs.re.kr; hyungjin.kim@stonybrook.edu

DNA replication is one of the most fundamental biological processes for the survival of an organism. Eukaryotic DNA replication is coordinated by specialized replisome machinery, including the Cdc45-MCM-GINS (CMG) helicase complex that unwinds the parental duplex DNA, and replicative polymerases that synthesize the daughter strand[1,2]. In addition, proliferating cell nuclear antigen (PCNA) is a DNA clamp that acts as a processivity factor to guide DNA polymerases and as a scaffold to coordinate the replication stress response that protects against DNA damage[3]. The DNA replication fork inevitably exposes single-stranded DNA (ssDNA) and is prone to damage that arises from various intrinsic and extrinsic barriers against fork progression[4]. Improper control of DNA replication stress results in the stalling and subsequent instability of replication forks, rendering DNA susceptible to nucleolytic degradation and breakage, triggering genome instability and tumor development[5]. Due to their hyper-proliferative nature, cancer cells are often selected through the loss of mechanisms to control DNA replication and exhibit elevated levels of replication stress[6]. Thus, exacerbating DNA replication stress in cancer cells has emerged as a new strategy to specifically kill cancer cells.

Cells have evolved sophisticated genome surveillance mechanisms to protect stalled replication forks and preserve fork integrity. The DNA replication stress response pathway, primarily driven by the intra-S checkpoint that activates ATR-dependent CHK1 phosphorylation, acts to inhibit cell cycle progression, stimulate DNA repair, modulate origin firing, and assist in the restart of stalled replication forks[7]. RPA-coated ssDNA, generated by the uncoupling of the CMG helicase and replicative polymerase activities, acts as a platform to recruit the ATR-binding partner ATRIP and stimulate ATR kinase activity through TOPBP1 at the 5′ ssDNA-double-stranded DNA (dsDNA) junction and also RPA-mediated ETAA1 recruitment, leading to CHK1 activation[8–12]. In addition, emerging evidence suggests that fork reversal plays a key role in protecting stressed replication forks[13]. This process involves regression of a stalled fork to form a four-way junction by the action of the RAD51 recombinase and several SWI/SNF family translocases such as SMARCAL1, ZRANB3, and HLTF[14,15]. This unique transaction protects stressed forks from degradation and acts as an intermediate for the repair and restart of stalled forks[16–18]. The tumor suppressor proteins BRCA1/2 and the Fanconi anemia (FA) protein FANCD2 are required for the protection of reversed forks by stabilizing the RAD51 filaments and preventing the regressed arm from nucleolytic degradation[19]. Many regulatory proteins fine-tune the steps of processing or protecting reversed forks, and deregulated fork remodeling has been associated with fork collapse, genome instability, and sensitivity or resistance to chemotherapy[20,21].

The fork protection complex (FPC), composed of TIMELESS (TIM) and TIPIN (Swi1 and Swi3 in *S. pombe*, and Tof1 and Csm3 in *S. cerevisiae*), and the ancillary proteins AND-1 and CLASPIN (CLSPN) stabilizes the replisome to ensure unperturbed fork progression[22–24]. The FPC acts as a scaffold to link the movements of the CMG helicase and polymerase, preventing the uncoupling of their activities and ensuring efficient replisome progression[25,26]. In addition, the FPC promotes the ATR-CHK1 checkpoint signaling during replication stress by stimulating the association of TIM-TIPIN and CLSPN to RPA-ssDNA at stalled forks, thereby facilitating CLSPN-mediated CHK1 phosphorylation by ATR[27,28]. TIM and TIPIN form an obligate heterodimer and have no known enzymatic function, suggesting that they play a structural role in supporting replisome integrity[29]. Loss of the FPC leads to defects in DNA replication and genome instability, indicating that maintaining structural integrity of the replisome is critical for preserving fork stability[30–32]. TIM is upregulated in a variety of cancers, implying that enhanced FPC activity may alleviate replication stress arising in tumors through oncogene activation[33]. However, the regulatory mechanism through which the FPC is controlled at active and stalled replication forks remains unclear.

We previously identified human SDE2 as a PCNA-associated protein required for counteracting replication-associated DNA damage[34]. SDE2 contains a ubiquitin-like (UBL) motif at its N-terminus that is cleaved off in a PCNA-dependent manner to release a C-terminal (Ct) SDE2[Ct] fragment (Fig. 1a). SDE2[Ct] contains a conserved SDE2 domain of unknown function at its N-terminus, while its C-terminal SAP domain mediates the association of SDE2[Ct] with chromatin. We further showed that chromatin-associated degradation of SDE2[Ct] by Arg/N-end rule-p97 ATPase proteolytic pathway is necessary for propagating the signaling of the DNA replication stress response at RPA-coated stalled forks in response to UVC damage[35]. These findings indicate that the dynamic control of SDE2 protein levels may modulate protein complexes and their activities at stressed forks. Although a role of SDE2 in DNA replication and stalled fork recovery is known, the mechanism by which this occurs is unknown[34]. Here, we demonstrate that SDE2 directly interacts with the FPC component TIM and promotes TIM stability and localization at replication forks. Consequently, loss of SDE2 or TIM compromises the integrity of the FPC, leading to defects in fork progression, stalled fork recovery, and checkpoint activation. Notably, SDE2 cooperates with TIM in protecting reversed forks from unrestricted nuclease activity, a role not previously associated with the FPC. We propose that SDE2 fulfills an essential role in active replication by promoting the association of the FPC with the replisome and participates together with TIM in the protection of stalled forks during fork reversal.

## Results

**SDE2 is localized at active DNA replication forks**. Based on our previous findings on the role of SDE2 in DNA replication, we asked whether SDE2 specifically associates with DNA replication forks, and if so, how it contributes to DNA replication and the stress response. To this end, we combined the proximity-ligation assay (PLA) with EdU labeling (also called in situ analysis of protein interactions at DNA replication forks, or SIRF) to quantitatively analyze the presence of proteins at newly synthesized DNA and validated the assay using PCNA as a control (Supplementary Fig. 1a)[36]. The PLA assay performed with SDE2 and biotin antibodies resulted in nuclear PLA foci, indicating the presence of endogenous SDE2 localized in close proximity to biotinylated EdU-labeled active DNA replication forks (Fig. 1b, c). The PLA signal is reduced to background levels in SDE2 knockdown cells, showing the specificity of the interaction. We could also visualize specific PLA foci between PCNA and GFP-tagged SDE2, further supporting its association with replication forks (Fig. 1d). The formation of SDE2-EdU PLA foci was dependent on the C-terminal SAP domain, which is required for DNA binding[34] (Supplementary Fig. 1b). Furthermore, isolation of proteins on nascent DNA (iPOND) coupled with thymidine-chase identified SDE2 associated with newly replicated DNA, similar to other known fork-associated proteins (Fig. 1e). In addition, we performed proximity biotinylation using an engineered peroxidase APEX2 fused to SDE2, for the identification of proteins associated with SDE2 in living cells. In this approach, proteins within a nanometer range of the bait protein are covalently tagged with biotin upon exposure to biotin-phenol (BP) and hydrogen peroxide ($H_2O_2$)[37] (Fig. 1f). We stably expressed Flag-tagged SDE2 fused to APEX2 at the C-terminus and

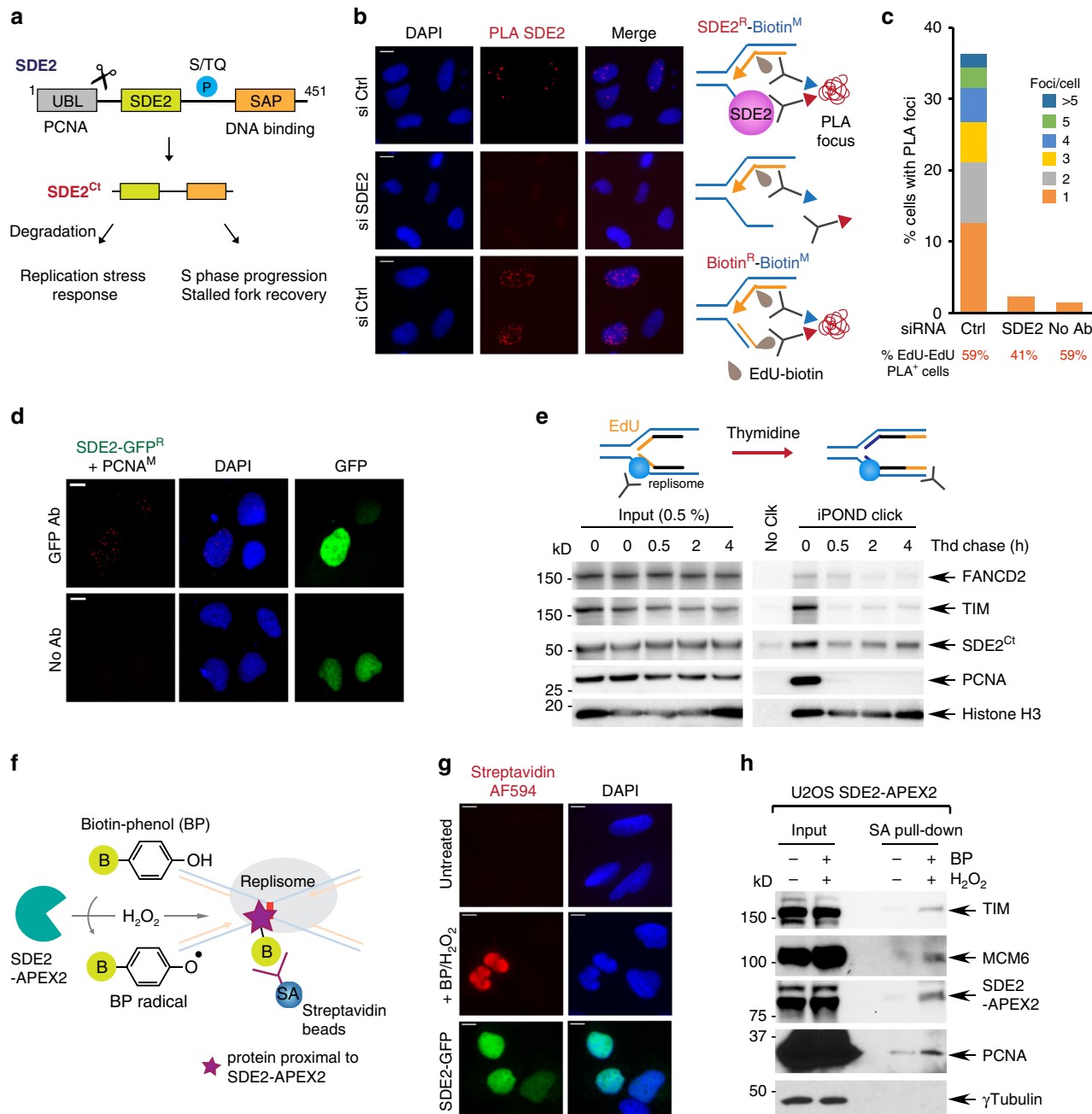

**Fig. 1 SDE2 is localized at replication forks. a** Human SDE2 domain structures and roles of the processed SDE2$^{Ct}$ in the replication stress response and replication fork integrity. **b** Representative images of SDE2:EdU-biotin PLA foci with schematic of the PLA design. The biotin:biotin PLA was used as a positive control to mark replicating cells. M: mouse, R: rabbit, primary antibodies used. Scale bar, 10 μm. **c** Quantification of cells positive for SDE2:EdU PLA foci and the number of foci per nuclei. A representative graph from two independent experiments is shown. More than 150 cells were scored per experimental condition. **d** PLA between SDE2-GFP and endogenous PCNA. Scale bar, 10 μm. **e** Western blot (WB) analysis of iPOND samples from EdU pulse and thymidine chase to reveal the proteins specifically associated with the replisome. **f** Schematic of the SDE2-APEX2 in situ proximity biotinylation to tag proteins proximal to SDE2. **g** Fluorescence imaging of SDE2-APEX2 labeling of nuclear proteins, detected by streptavidin-conjugated Alexa Fluor (AF) 594. Nuclear localization of SDE2-GFP was confirmed by GFP fluorescence. Scale bar, 10 μm. **h** WB analysis of biotinylated proteins by proximity labeling. U2OS cells stably expressing SDE2-APEX2 were incubated with BP and H$_2$O$_2$, and biotinylated proteins were enriched by streptavidin pull-down.

confirmed the biotinylation of endogenous proteins upon BP and H$_2$O$_2$ treatment (Supplementary Fig. 1c, d). Fluorescence staining with fluorescent streptavidin conjugates showed that the biotinylated proteins are confined within the nucleus (Fig. 1g). Interestingly, specific enrichment of SDE2-associated proteins by streptavidin pull-down revealed that known replisome components such as PCNA and MCM6, along with TIM, a core component of the FPC, are associated with SDE2 (Fig. 1h).

Together, these results suggest that SDE2 is present at sites of DNA replication and interacts with the replication machinery.

**SDE2 directly interacts with the C-terminus of TIM.** The observation that TIM is in close proximity to SDE2 at replication forks prompted us to determine whether SDE2 physically interacts with TIM. We were indeed able to co-immunoprecipitate endogenous TIM with an SDE2 antibody (Fig. 2a). Flag-tagged

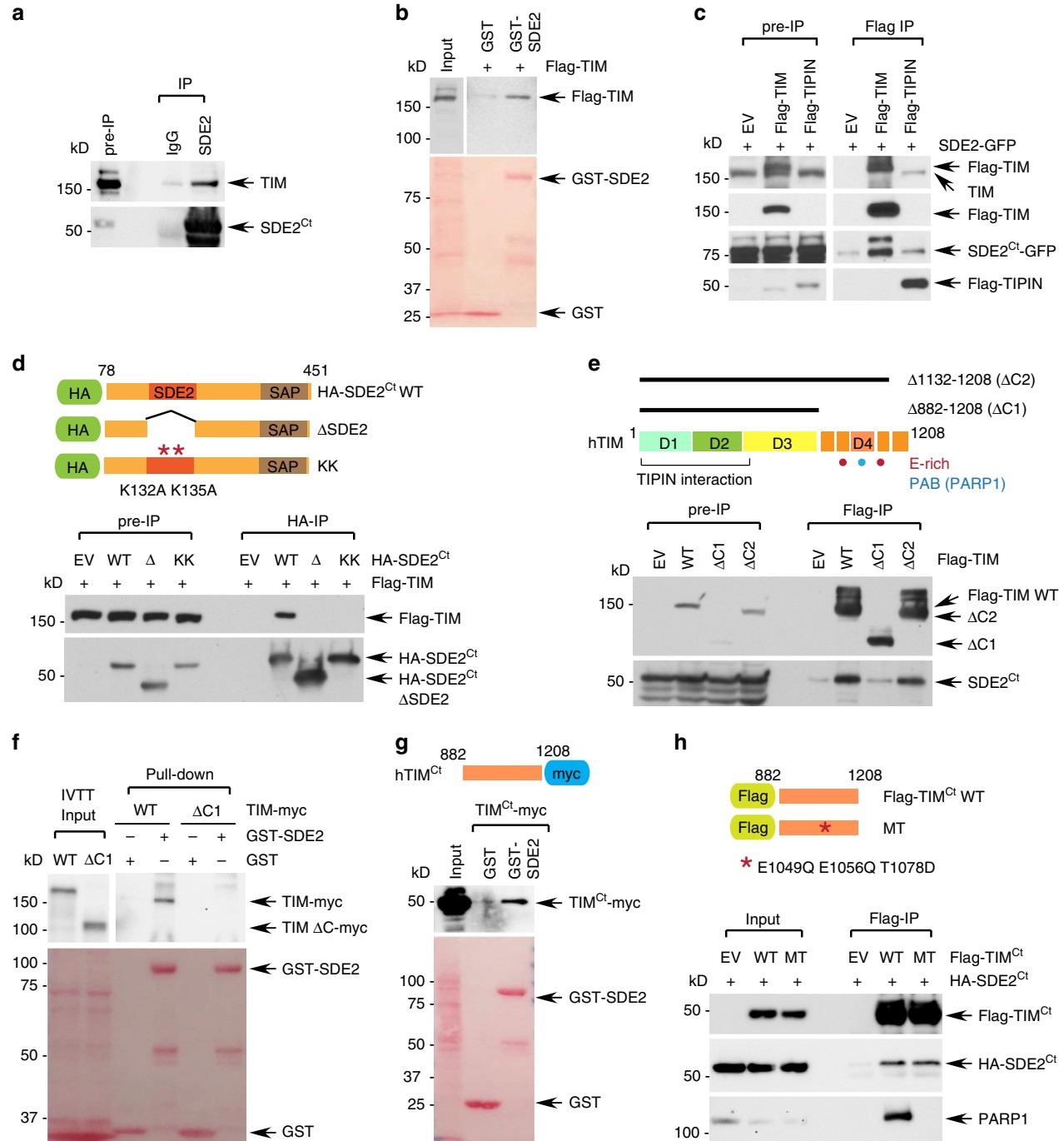

**Fig. 2 SDE2 directly interacts with the C-terminus of TIM. a** Co-immunoprecipitation (IP) of both endogenous SDE2 and TIM in U2OS cells using an SDE2 antibody (vs. rabbit IgG control). **b** GST pull-down of 293T cell lysates expressing Flag-TIM with purified GST or GST-SDE2. **c** Anti-Flag co-IP of Flag-TIM or Flag-TIPIN with SDE2-GFP in 293T cells. EV: empty vector. **d** Anti-HA co-IP of Flag-TIM with HA-SDE2^Ct wild-type (WT), ΔSDE2, or KK (K132A/K135A) mutant in 293T cells. Schematic of HA-SDE2 construct variants is shown above. **e** Anti-Flag co-IP of Flag-TIM WT, Δ882-1208 (ΔC1), or Δ1132-1208 (ΔC2) with endogenous SDE2. Schematic of human TIM is shown above. D1-D4 indicates four domains classified as previously described. The TIM C-terminus referred in this study is indicated in orange. Red dot: Glu-rich regions. Blue dot: PARP1-interacting region, PAB, or D4. **f** GST pull-down of in vitro transcribed and translated (IVTT) TIM-myc WT or ΔC1 mutant with purified GST or GST-SDE2. **g** GST pull-down of 293T cell lysates expressing TIM C-terminus (Ct; aa882-1208)-myc with purified GST or GST-SDE2. **h** Anti-Flag co-IP of Flag-TIM^Ct WT or PARP1-binding mutant (MT: E1049Q/E1056Q/T1078D) either with HA-SDE2^Ct or endogenous PARP1.

SDE2 expressed in U2OS cells also interacted with endogenous TIM (Supplementary Fig. 2a). Recombinant GST-tagged SDE2 was able to pull down Flag-tagged TIM, showing that the interaction between the two proteins is direct (Fig. 2b). SDE2-GFP was co-immunoprecipitated not only by Flag-TIM, but also by Flag-TIPIN, the obligate partner of TIM in the FPC, suggesting that SDE2 is associated with the TIM-TIPIN complex in cells (Fig. 2c). Importantly, deletion of the SDE2 domain (ΔSDE2) or point mutations of conserved lysine residues within the SDE2 domain abrogated the interaction with TIM, indicating that the

SDE2 domain is crucial for the interaction (Fig. 2d). By contrast, the SDE2 ΔSAP mutant maintained its interaction with TIM, indicating that the DNA-binding role of the SAP domain is not required for SDE2-TIM interaction (Supplementary Fig. 2b).

Previous studies have defined four conserved regions of mouse Timeless, denoted D1-D4, and showed that the C-terminus of TIM that includes D4, harbors several Glu (E)-rich regions[38] (Fig. 2e). Deletion mutagenesis of TIM revealed that the C-terminal area encompassing the E-rich regions (aa882-1132; ΔC1) is necessary for the interaction with SDE2 (Fig. 2e). Indeed, the TIM ΔC1 mutant failed to interact with recombinant SDE2 in vitro (Fig. 2f). In addition, the TIM C-terminus either expressed in cells or as an in vitro translated protein, was sufficient to interact with recombinant SDE2, suggesting that the C-terminus of TIM directly interacts with SDE2 (Fig. 2g; Supplementary Fig. 2c). TIPIN, the obligate partner of TIM, is known to directly interact with the N-terminus (D1 and D2) of TIM[31,39]. We found that the TIM Δ1-603 (ΔN) mutant, which fails to bind to TIPIN, still interacts with SDE2, indicating that the interactions of TIM with TIPIN and with SDE2 are mediated by distinct regions of TIM (Supplementary Fig. 2d, e). Previous structural studies demonstrated that TIM directly interacts with PARP1 via TIM's C-terminal PAB (PARP1-binding) domain, located within the D4 region, and identified several residues including E1049, E1056, and T1078 to be essential for PARP1 interaction[40,41]. However, the TIM C-terminal point mutant that fails to bind to PARP1 still retained its interaction with SDE2, suggesting that SDE2 binds to TIM via a region distinct from the PAB domain (Fig. 2h). Together, we conclude that SDE2 directly interacts with the C-terminus of TIM via its conserved SDE2 domain, constituting an SDE2-TIM-TIPIN complex within the FPC.

**SDE2 is required for the stability and localization of TIM at replication forks.** Given that SDE2 directly binds to TIM, we then asked whether SDE2 has the potential to affect the FPC stability and function through its interaction with TIM. Cell synchronization analysis revealed that, as cells progress toward S phase, the levels of both SDE2 and TIM increase specifically in the chromatin-enriched (P), but not in the soluble (S) fraction, indicating that their levels at DNA might be coordinated (Supplementary Fig. 3a). Notably, SDE2 depletion using two independent siRNAs led to a decrease in TIM levels specifically in the P fraction, and addition of the proteasome inhibitor MG132 restored TIM in the P fraction, showing that SDE2 prevents proteasomal degradation of TIM associated with DNA (Fig. 3a, b). Indeed, SDE2 depletion caused accelerated TIM degradation in the P fraction, indicating that turnover of chromatin-associated TIM is controlled by SDE2 (Fig. 3c). The TIM ΔC1 mutant that fails to interact with SDE2 was also degraded more rapidly than TIM (Supplementary Fig. 3b). Exogenous expression of WT SDE2, but not the ΔSDE2 or KK mutants, was sufficient to increase levels of TIM, indicating that the interaction of SDE2 with TIM promotes TIM stability (Fig. 3d; Supplementary Fig. 3c, d). As the obligate binding partner of TIM, TIPIN depletion led to destabilization of TIM in both S and P fractions as well as in whole-cell lysates (Fig. 3e). By contrast, SDE2 was responsible for sustaining TIM levels mainly in the P fraction, suggesting that SDE2 regulates TIM specifically at replication forks (Fig. 3e). We next checked whether the destabilization effect of SDE2 knockdown on TIM levels in the chromatin is translated into a localization defect at nascent DNA. We therefore used the PLA assay to assess the presence of endogenous TIM at EdU-labeled replication forks. Knockdown of SDE2 resulted in decreased numbers of TIM PLA-positive cells and number of foci per nucleus,

providing additional evidence that the association of TIM with the replisome is disrupted in the absence of SDE2 (Fig. 3f; Supplementary Fig. 3e). We next determined whether this role of SDE2 in TIM regulation occurs in the context of the FPC. To this end, we devised a PLA assay between MCM6 and EdU as a way to assess the coupling activity of the CMG helicase and polymerase. Treatment with HU resulted in a reduction of MCM6-EdU PLA-positive cells, representing fork stalling and ssDNA accumulation (Supplementary Fig. 3f). Similarly, knocking down SDE2 or TIM led to a significant decrease in cells positive for the MCM6-EdU PLA foci, indicating that the replisome movement is uncoupled (Fig. 3g). Together, these results suggest that SDE2 promotes the stability of TIM and thus the integrity of the FPC at replication forks, as required for replisome progression.

**SDE2 and TIM are required for efficient fork progression and stalled fork recovery.** Our results thus far predict that SDE2 deficiency would phenocopy TIM deficiency in DNA replication fork integrity and checkpoint activation, both of which are functions of the FPC. To understand the role of TIM and its regulator SDE2 in DNA replication, we employed a single-molecule DNA combing assay to monitor individual DNA replication tracks. Knocking down SDE2 or TIM in U2OS cells resulted in a significant shortening of replication tracks as assessed both by the distribution of fiber lengths and their median values, indicating that SDE2 and TIM are required for efficient fork progression (Fig. 4a, b). A similar result was observed in non-transformed BJ-TERT cells (Supplementary Fig. 4a). Moreover, the frequency of asymmetric replication tracks was substantially increased without SDE2 or TIM, indicating that forks are frequently stalled (Fig. 4c). Lack of SDE2 or TIM also resulted in shorter inter-origin distance, indicative of an increase in new origin firing and fork slow-down under stress (Fig. 4d). Cell cycle analysis revealed a significant decrease in the number of replicating cells upon SDE2 or TIM depletion, especially in the late S phase, indicating that S phase progression is impaired (Supplementary Fig. 4b, c). Co-depletion of SDE2 and TIM did not further exacerbate the replication defect of single knockdown, supporting an epistatic relationship between the two proteins (Fig. 4e; Supplementary Fig. 4d). To determine whether SDE2 and TIM also share a role in fork recovery during replication stress, we exposed cells to hydroxyurea (HU) to stall replication forks and quantified the number of forks that were able to resume replication upon removal of HU (Fig. 4f). SDE2 or TIM knockdown resulted in a defect in stalled fork recovery, as measured by both fork recovery efficiency and IdU track lengths, with a severity similar to FANCD2 knockdown, and fork recovery was not further compromised by co-depletion of SDE2 and TIM, indicating that SDE2 and TIM work together in fork recovery (Fig. 4f; Supplementary Fig. 4e–g).

**SDE2 and TIM are required for efficient checkpoint activation.** In addition to its role in stabilizing active replication forks, the FPC plays a key role in promoting the replication checkpoint. Specifically, the interaction of TIPIN with RPA is known to stabilize the FPC on RPA coated-ssDNA to potentiate CLSPN-mediated CHK1 phosphorylation by ATR[27]. We thus determined whether SDE2 is necessary for the DNA damage response and checkpoint pathways. Knockdown of SDE2 resulted in the formation of DNA breaks, as shown by the increased tail length in alkaline DNA comet assays upon HU treatment or UVC irradiation (Fig. 5a; Supplementary Fig. 5a). This is accompanied by both elevated levels of γH2AX specifically in EdU-positive cells and BrdU-labeled ssDNA following HU treatment, indicating increased replication-associated DNA damage in the absence of

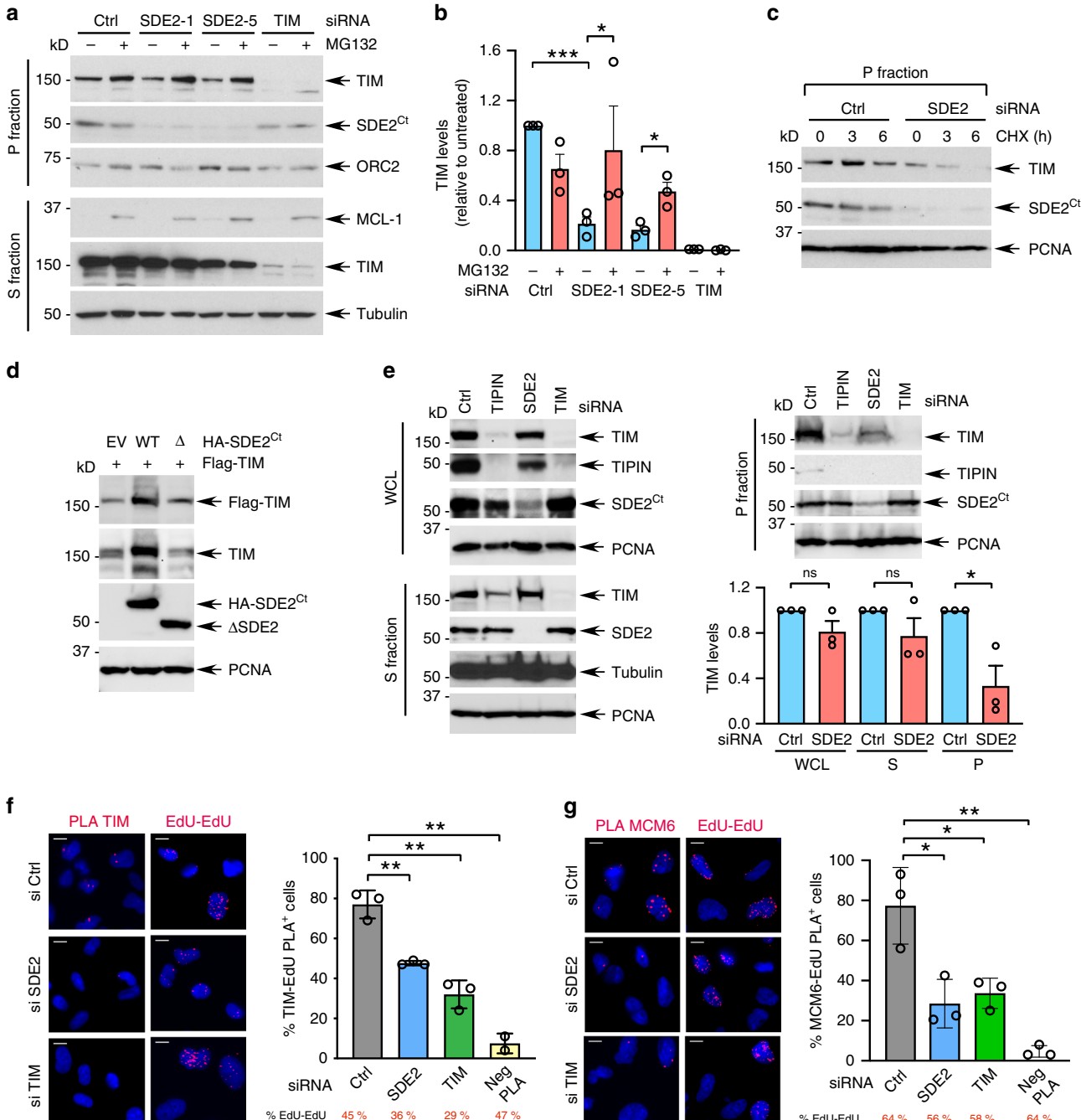

**Fig. 3 SDE2 promotes the stability and localization of TIM at replication forks. a** Endogenous TIM levels in S and P fractions with SDE2 depleted. U2OS cells transfected with the indicated siRNAs were fractionated into S and P fractions and analyzed by WB. Where indicated, cells were treated with 20 μM MG132 for 6 h before harvest. **b** Quantification of TIM levels in the P fraction (n = 3 biologically independent experiments, mean ± SEM, *P < 0.05, Student's t-test). **c** Degradation of TIM in the P fraction. siRNA-transfected U2OS cells were treated with 100 μg/mL cycloheximide (CHX) for the indicated times, and the P fraction was analyzed by WB. **d** WB analysis of 293T cells co-transfected with Flag-TIM and either HA-SDE2^Ct WT or ΔSDE2 mutant. **e** WB analysis of whole-cell lysate (WCL), S, and P fractions derived from U2OS cells transfected with the indicated siRNAs. TIM levels in each fraction were quantified (n = 3 biologically independent experiments, mean ± SEM, *P < 0.05, Student's t-test, ns, not significant). Data were normalized to the corresponding percentage of EdU-EdU PLA-positive cells. **f** Left: TIM:EdU PLA from U2OS cells transfected with the indicated siRNAs. Scale bar, 10 μm. Right: quantification of cells positive for TIM:EdU PLA foci out of EdU+ cells (>300 cells per condition, n = 3 biologically independent experiments, mean ± SD, **P < 0.01, Student's t-test). Neg PLA indicates the omission of the Biotin antibody. **g** Left: MCM6:EdU PLA foci from U2OS cells transfected with the indicated siRNAs. Scale bar, 10 μm. Right: quantification of cells positive for MCM6:EdU PLA foci out of EdU+ cells (>300 cells per condition, n = 3 biologically independent experiments, mean ± SD, **P < 0.01, *P < 0.05, Student's t-test). Data were normalized to the corresponding percentage of EdU-EdU PLA-positive cells.

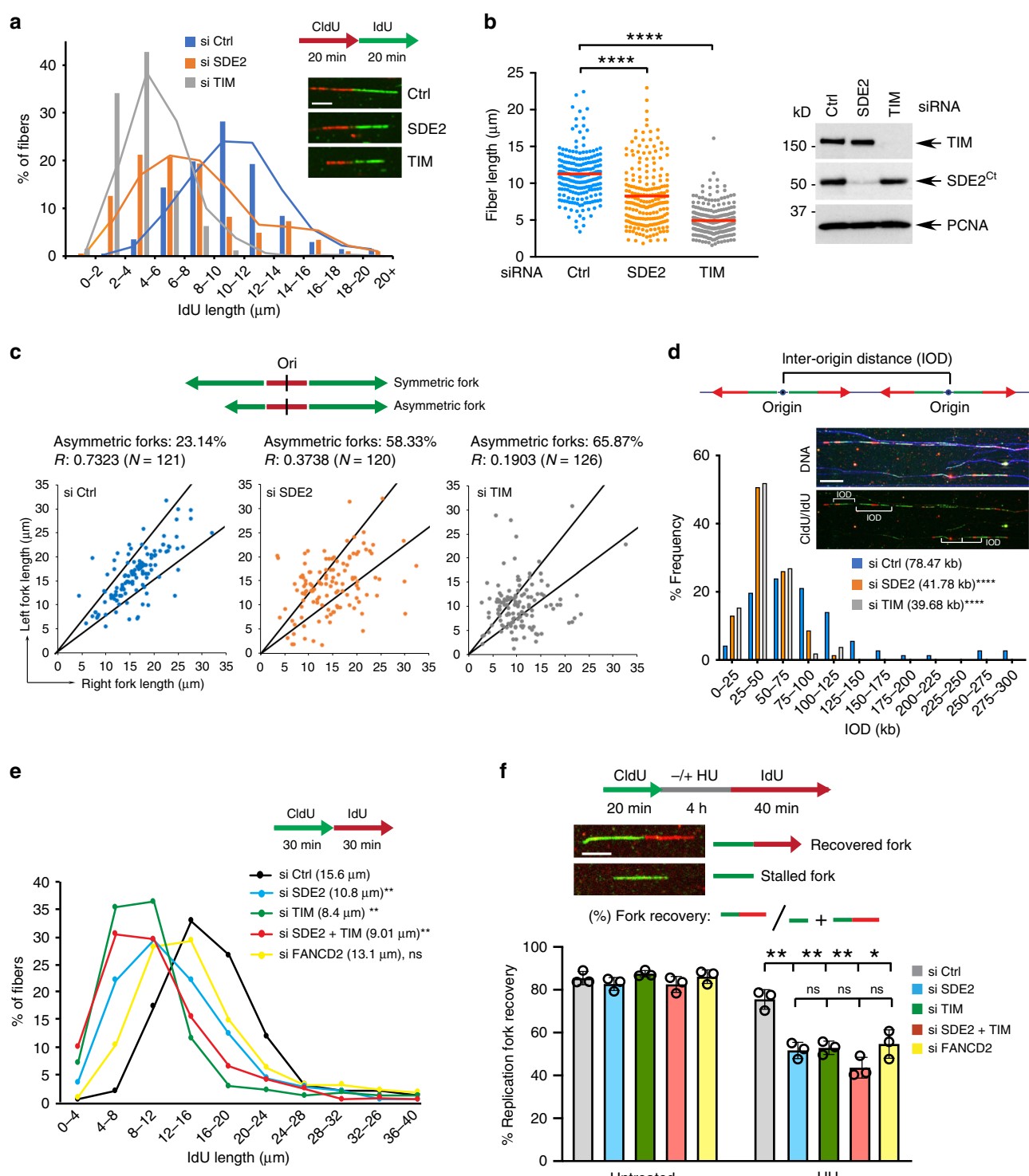

SDE2 or TIM (Fig. 5b; Supplementary Fig. 5b, c). As expected, CHK1 phosphorylation was significantly impaired upon SDE2 or TIM knockdown following HU treatment (Fig. 5c, d). Similar results were observed upon UVC irradiation in cells with three independent SDE2 siRNAs, and upon camptothecin (CPT) treatment (Supplementary Fig. 5d, e). The ATR-CHK1 activation cascade is stimulated by both TOPBP1, which works at the 5′ ssDNA-dsDNA junctions of stalled forks, and ETAA1, which is primarily recruited to the ssDNA-RPA platform for ATR interaction[7]. Previous studies have shown that, in contrast to TOPBP1, ETAA1 depletion has no noticeable effect on CHK1

phosphorylation, indicating that the two activators work in parallel to regulate a subset of ATR targets[11,12]. Notably, the effect of SDE2 or TIM depletion in preventing CHK1 phosphorylation was as just pronounced as the effect from TOPBP1 depletion, while ETAA1 knockdown did not impair CHK1 activation to the same extent (Fig. 5e). This provides support for the role of the FPC and SDE2 in stimulating ATR-mediated CHK1 phosphorylation via the RAD9-RAD1-HUS1 (9-1-1) clamp-TOPBP1 pathway when enriched at stalled forks. Importantly, exogenous expression of TIM was able to partially restore pCHK1 levels following SDE2 depletion, indicating that reduced TIM levels

**Fig. 4 SDE2 and TIM promote fork progression and stalled fork recovery. a** DNA combing analysis. Distributions of the IdU track length were determined on CldU:IdU double-labeled DNA fibers from siRNA-transfected U2OS cells ($n = 2$, a representative experiment is shown). **b** Left: dot plot of the IdU track length, $n = 2$ biologically independent experiments (****$P < 0.0001$, Mann–Whitney test). Red bars indicate the median value of at least 200 tracks per experimental condition. Scale bar, 5 μm. Right: WB to confirm knockdown. **c** Fork asymmetry analysis of siRNA-transfected U2OS cells. Sister forks from initiation events (green-red-green tracks only) are plotted from two technical replicates. The central area between the lines delimits a variation of <25% in sister forks length. R represents the linear correlation coefficient. **d** Distribution of inter-origin distance from siRNA-transfected U2OS cells. Upper panels show DNA counterstaining in blue and DNA tracks originated from CldU (green) and subsequently IdU (red) labeling ($n = 2$ biologically independent experiments, median value from at least 50 fibers, ****$P < 0.0001$, the two-tailed Mann–Whitney). Scale bar, 50 μm. **e** Distributions of the IdU track length from U2OS cells transfected with individual siRNA or in combination. FANCD2 knockdown, which does not impair fork progression, serves as a negative control. The median value of at least 200 tracks per experimental condition is indicated ($n = 3$, **$P < 0.01$, Student's $t$-test). **f** Top: DNA combing protocol with a schematic of different replication dynamics. Fork recovery is measured as the portion of the number of green-red fiber tracks divided by that of green-only and green-red fibers. Scale bar, 5 μm. Bottom: replication fork restart after HU-mediated fork stalling in U2OS cells transfected with the indicated siRNAs ($n = 3$ biologically independent experiments, mean ± SD, **$P < 0.01$, *$P < 0.05$, Student's $t$-test). FANCD2 knockdown serves as a control for a defect in fork recovery.

caused by SDE2 deficiency are at least partly responsible for the compromised pCHK1 levels (Fig. 5f). Moreover, recruitment of CLSPN to chromatin upon HU treatment was impaired in both SDE2- and TIM-depleted cells, further supporting the role of SDE2 and TIM in CHK1 activation via the engagement of CLSPN in the FPC at stalled forks (Fig. 5g). Intriguingly, a previous study demonstrated that deficiency of TIM causes ssDNA accumulation and an increased dependence on ATR for continuous DNA synthesis in unchallenged conditions[42]. Thus, the FPC promoting ATR-pCHK1 pathway may serve as a localized adaptor function under replication stress that occurs specifically at sites of stalled forks. Overall, these results suggest that SDE2 cooperates with TIM to control the ATR-CHK1 checkpoint.

**SDE2 and TIM are required for protecting reversed forks.** Stressed forks are known to be subjected to nucleolytic degradation, leading to impaired fork integrity and stalled fork recovery[13]. To determine the possible role of the FPC in protecting stressed forks, we treated cells with a low dose of HU overnight and released them into fresh medium to allow for stalled fork recovery. Knockdown of SDE2 or TIM resulted in a dramatic intensification of RPA32 phosphorylation at S4/S8, indicative of hyper-resection of DNA repair intermediates and DNA break formation[43,44]. By contrast, WT cells exhibited minimal levels of phosphorylated RPA, indicating that stressed forks are not properly resolved in the absence of SDE2 or TIM (Fig. 6a; Supplementary Fig. 6a). Re-expression of TIM was sufficient to suppress the elevated pRPA levels from TIM knockdown, confirming phenotype specificity (Supplementary Fig. 6b). We also observed an increased number of cells positive for pRPA32 S4/S8 by both flow cytometry and immunofluorescence (Fig. 6b, c). Furthermore, pRPA signals dramatically increased at stalled replication forks triggered by HU in SDE2- or TIM-depleted cells, as revealed by iPOND (Fig. 6d).

To understand the mechanism by which SDE2 and TIM protect stressed forks, we monitored fork resection by DNA combing analysis. Cells depleted of SDE2 or TIM and exposed to HU exhibited excessive fork degradation, evidenced both by shortening of replication track lengths and by decreased IdU/CldU ratios (Fig. 6e, f). Co-depletion of SDE2 and TIM did not further exacerbate the fork resection of single knockdowns, suggesting that they work in the same fork protection pathway (Supplementary Fig. 6c). Recent evidence showed that stalled forks frequently undergo fork reversal, when the two DNA strands reverse their course and anneal to each other to provide a favorable DNA substrate that can promote fork protection and restart[16–18]. To test whether TIM and SDE2 protect reversed forks, we co-depleted SMARCAL1, a fork remodeler that

generates reversed forks[15]. Depletion of SMARCAL1 indeed rescued cells from excessive fork degradation in the absence of SDE2 and TIM, indicating that SDE2 and TIM protect reversed forks from degradation (Fig. 6e, f; Supplementary Fig. 6d). To further understand the mechanism of TIM-mediated fork protection, we depleted individual nucleases known to work on the processing of stalled forks and determined whether TIM protects forks from nucleolytic degradation (Supplementary Fig. 6e). Interestingly, co-knockdown of TIM with MRE11, but not with CtIP, DNA2, or EXO1, led to a reduction of the pRPA S4/S8 levels elevated by TIM depletion (Fig. 6g; Supplementary Fig. 6f). Similarly, inhibition of the MRE11 nuclease activity by the small molecule mirin attenuated fork resection in TIM-depleted cells, indicating that TIM protects reversed forks from MRE11-mediated nucleolytic degradation (Fig. 6h).

Of the multiple fork protection pathways that have been identified, BRCA2 is known to restrict MRE11 activity by stabilizing RAD51 nucleofilaments at reversed forks[17,18]. Although not always statistically significant, we observed a trend of worsened fork resection in BRCA2 and TIM double knockdown cells (Supplementary Fig. 6g). We faced challenges acquiring an adequate number of fiber tracks for analysis from these cells, indicating that replication fork integrity was severely compromised upon acute knockdown of BRCA2 and TIM. In line with this, simultaneous knockdown of both proteins resulted in nearly complete inhibition of cell growth, indicating a synergistic effect (Supplementary Fig. 6h). Therefore, TIM and BRCA2 may work in parallel pathways that converge on regulating MRE11 activity to protect reversed forks from degradation. Together, these results support the role of SDE2 and TIM in protecting reversed forks from nucleolytic degradation and identify the FPC as a new component of the fork protection mechanism at stalled forks.

**SDE2-TIM interaction controls DNA replication fork integrity and fork protection.** Our study thus far points to a regulatory role for SDE2 in controlling the integrity of TIM at replication forks and its associated function in the protection of damaged forks. To gain further insight into the underlying mechanism, we performed a structure-function analysis using SDE2 knockdown cells complemented with siRNA-resistant WT SDE2 or ΔSDE2 domain deletion mutant. In this Retro-X Tet-One system, doxycycline (dox) induces the expression of SDE2 cDNA under the $P_{TRE3GS}$ promoter via the Tet-On 3 G transactivator[45] (Fig. 7a). We confirmed the induction of exogenous SDE2 proteins at near-physiological levels in a dox-dependent manner, while endogenous SDE2 was successfully depleted. Using these cell lines, we investigated whether the SDE2-TIM interaction is essential for

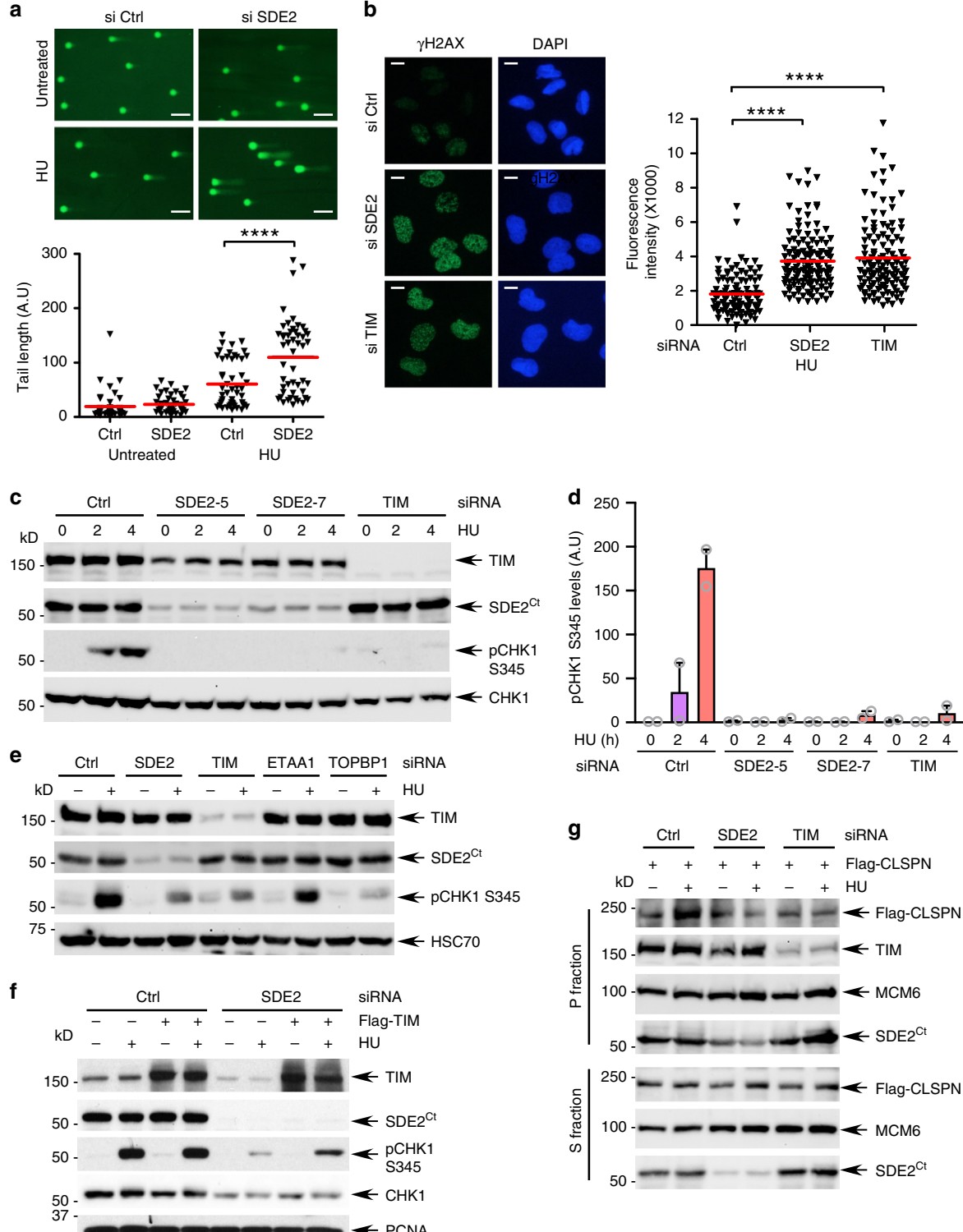

**Fig. 5 SDE2 promotes CHK1 activation via TIM and CLSPN. a** Top: representative images of DNA comets after treatment with 2 mM HU for 4 h in siRNA-transfected U2OS cells ($n = 2$ biologically independent experiments). Scale bar, 50 μm. Bottom: quantification of DNA comet tail lengths. Red bars represent the median (>50 nuclei per sample, ****$P < 0.0001$, Mann–Whitney test). **b** Left: representative images of γH2AX staining after treatment with 2 mM HU for 4 h in siRNA-transfected U2OS cells ($n = 2$ biologically independent experiments). Scale bar, 10 μm. Right: Quantification of γH2AX intensity. Red bars represent the median (>100 nuclei per sample, ****$P < 0.0001$, Mann–Whitney test). **c** CHK1 phosphorylation after treatment of 2 mM HU for the indicated times in siRNA-transfected U2OS cells. **d** Quantification of pCHK1 levels normalized by total CHK1. Mean from two biologically independent experiments is shown. **e** Comparison of pCHK1 induction in U2OS cells transfected with the indicated siRNAs. **f** Rescue of pCHK1 induction in U2OS cells transfected with SDE2 siRNA followed by Flag-TIM expression. **g** Impaired Flag-CLSPN recruitment to the chromatin-enriched P fraction in subcellular-fractionated U2OS cells transfected with SDE2 or TIM siRNA, followed by Flag-CLSPN expression.

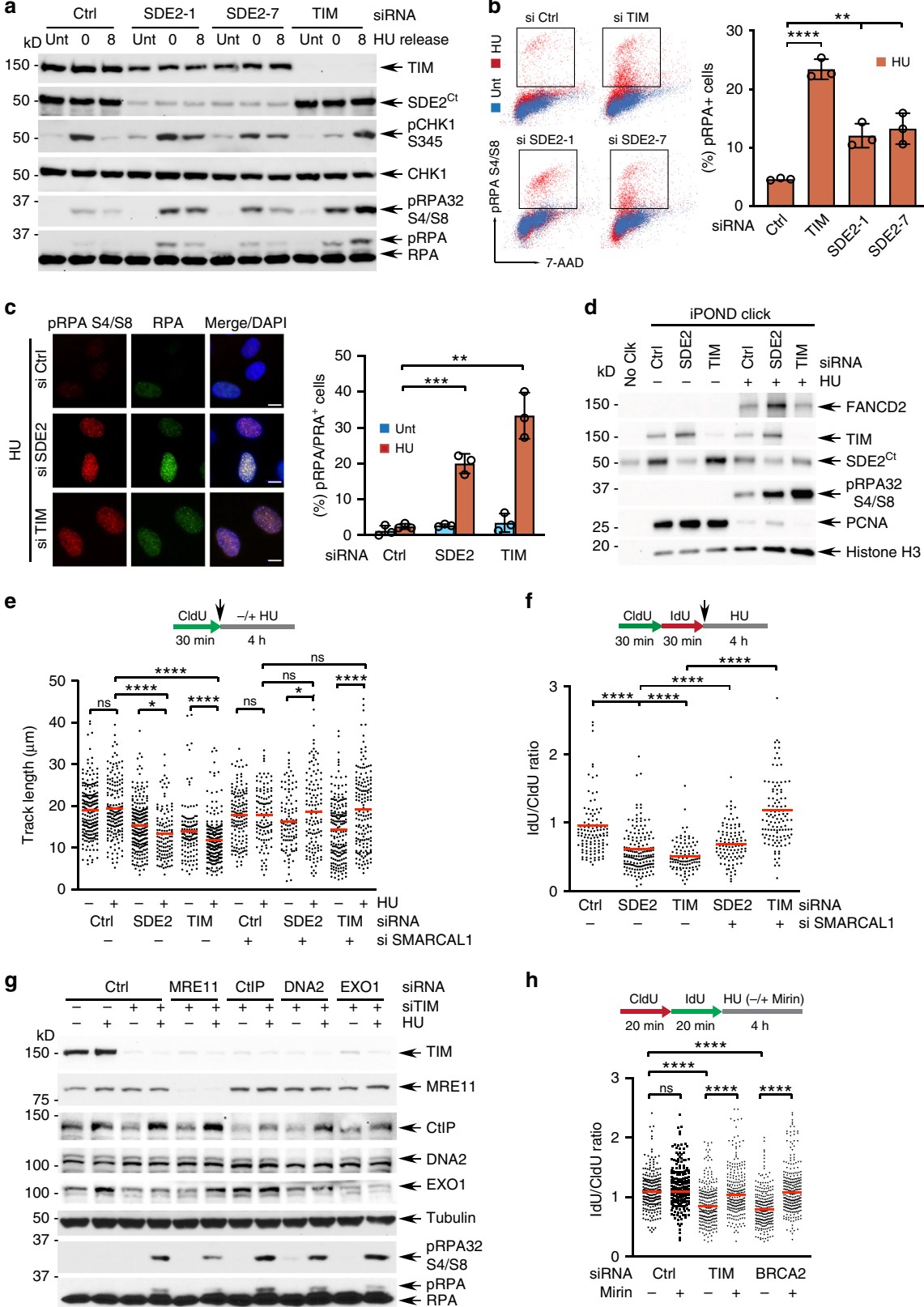

TIM function in preserving fork integrity. Subcellular fractionation experiments showed that SDE2 WT complementation is able to rescue decreased TIM levels in the chromatin after SDE2 knockdown, while ΔSDE2 domain deletion mutant fails to do so (Fig. 7b; Supplementary Fig. 7a). Accordingly, a TIM-EdU PLA assay revealed that the ΔSDE2 mutant was not able to sustain

TIM localization at EdU-labeled forks, confirming that SDE2 recruits TIM and promotes its localization to replication forks (Fig. 7c; Supplementary Fig. 7b). Furthermore, cells reconstituted with the ΔSDE2 mutant were not able to restore replication track lengths in comparison to WT SDE2, suggesting that the SDE2-TIM interaction is necessary for efficient fork progression

**Fig. 6 SDE2 and TIM are required for protecting reversed forks and counteracting nuclease activity. a** RPA32 S4/S8 phosphorylation in siRNA-transfected U2OS cells. Cells were either untreated (Unt) or treated with 250 µM HU for 20 h and released into fresh medium for the indicated times before harvest. See Fig. S6A for quantification. **b** Left: representative images of flow cytometry analysis of siRNA-transfected U2OS cells either untreated or treated with 250 µM HU for 18 h, and stained with anti-pRPA32 S4/S8 antibody. Right: quantification of pRPA32+ cells ($n = 3$ biologically independent experiments, mean ± SD, ****$P < 0.0001$, **$P < 0.01$, Student's t-test). **c** Left: representative images of pRPA32 S4/S8 immunofluorescence from siRNA-transfected U2OS cells treated with 250 µM HU for 16 h. Scale bar, 10 µm. Right: quantification of pRPA32 S4/S8+ cells ($n = 3$ biologically independent experiments, mean ± SD, ***$P < 0.001$, **$P < 0.01$, Student's t-test). **d** pRPA32 S4/S8 levels of the iPOND samples from siRNA-transfected U2OS cells treated with 2 mM HU for 4 h. **e** Dot plot of the CldU track length of siRNA-transfected U2OS cells untreated or treated with 2 mM HU for 4 h. A representative result from two independent experiments is shown (at least 150 tracks per condition, ****$P < 0.0001$, Mann–Whitney). **f** Dot plot of IdU/CldU ratios of U2OS cells co-depleted of SDE2 or TIM and SMARCAL1. A representative result from two independent experiments is shown (at least 200 track ratios per condition, ****$P < 0.0001$, Mann–Whitney). **g** Western blot analysis as Fig. 6a showing the rescue of increased pRPA32 S4/S8 levels by knockdown of MRE11 in TIM-depleted U2OS cells. For quantification, see Supplementary Fig. 6f. **h** Dot plot of DNA fiber IdU/CldU track length ratios from siRNA-transfected U2OS cells treated with 4 mM HU, and co-treated with 50 µM mirin or DMSO ($n = 2$ biologically independent experiments, at least 230 track ratios per condition, ****$P < 0.0001$, Mann–Whitney). BRCA2 knockdown cells serve as a positive control for the restoration of fork protection by the inhibition of MRE11 activity.

(Fig. 7d; Supplementary Fig. 7c). Similarly, the fork resection under HU-induced replication stress was significantly more pronounced in cells reconstituted with the SDE2 mutant (Fig. 7e). In addition, the TIM ΔC1 mutant that cannot interact with SDE2 failed to rescue the fork resection defect in TIM-depleted cells, indicating that the SDE2-TIM interaction is required for the protection of damaged forks (Supplementary Fig. 7d). Together, these results suggest that the SDE2-TIM interaction is essential for the FPC to carry out its roles in both DNA replication and protection of stalled forks from over-resection.

## Discussion

**SDE2 as a new regulatory component of the FPC.** The DNA replication machinery, known as the replisome, must coordinate multiple transactions during DNA replication, which necessitates stable association of its components with chromatin and its associated proteins. The FPC is known to interact with many components of the replisome, including MCM helicase subunits, the replicative polymerases δ and ε, and the ssDNA-binding factor RPA[25]. Yeast and metazoan studies have revealed that the FPC associates with replication origins at the onset of S phase and traverses along with the replisome during S phase progression to tether the helicase-polymerase activity and coordinate leading- and lagging-strand synthesis[30,46]. It also interacts with the structural maintenance of chromosomes (SMC) subunits of the cohesion complex to promote the establishment of sister chromatid cohesion (SCC) during DNA replication[30].

Despite its critical role as a platform to stabilize the replication fork and facilitate checkpoint signaling, it remains unclear how the FPC is controlled. We discovered that SDE2, a PCNA-associated genome surveillance protein located at replication forks, is required for the functional integrity of the FPC. We demonstrated that the conserved SDE2 domain is necessary for SDE2 to interact with the C-terminus of TIM, which promotes the stability and localization of TIM at replication forks. Using several independent approaches, we showed that SDE2 is localized to active replication forks, and we propose that the DNA-binding property of SDE2 via its SAP domain allows the FPC to stably associate with chromatin and guide its progression alongside the replisome (Fig. 7f). Indeed, loss of SDE2 or its interaction with TIM phenocopies TIM deficiency, leading to extensive ssDNA formation, impaired fork progression, and defects in stalled fork recovery. Similarly, deficiency of And-1 in chicken DT40 cells causes fork speed slow-down and ssDNA accumulation via its HMG DNA-binding domain, further supporting the role of the FPC acting as a scaffold at replication forks[47]. Interestingly, unlike TIPIN that constitutes an obligate

binding partner of TIM, SDE2 appears to play a regulatory role in modulating the stability of TIM predominantly on chromatin, and TIM depletion does not lead to destabilization of SDE2. This indicates that SDE2 may preferentially interact with and control the activity of TIM engaged in the replisome at the site of DNA replication. Since we observe the SDE2-TIM-TIPIN trimer and that the depletion of SDE2 causes uncoupling of replisome activity, SDE2 is expected to regulate TIM within the FPC. However, we do not exclude the possibility that SDE2 might regulate other TIM-specific functions. Indeed, TIM activity independent of the TIM-TIPIN heterodimer has been reported in promoting the SCC establishment[48]. The mechanism through which TIM becomes unstable in the absence of SDE2 is not known. The C-terminus of TIM may harbor an undiscovered degron that initiates chromatin-specific degradation signaling in the absence of SDE2, and replisome-associated ubiquitin E3 ligases may be involved. Additional investigation on the mechanism through which TIM stability is maintained cooperatively by TIPIN and SDE2 is an important future study to better appreciate how the functional integrity of the FPC is preserved at replication forks.

**Roles of TIM associated with fork protection during fork reversal.** While TIPIN is known to directly interact with RPA to transmit the signaling of ssDNA accumulation and activate CHK1, the precise function of the FPC in maintaining the integrity of stressed forks remains unclear. We show that TIM and its regulatory partner SDE2 are required for counteracting the excessive resection of reversed forks under replication stress (Fig. 7f). Emerging evidence indicates that a stressed fork experiences dynamic remodeling to reverse its course, which promotes fork stabilization and helps repair or bypass DNA damage. While many nucleases and DNA remodelers are involved, the fate of the replisome during the fork reversal is poorly understood. A previous iPOND analysis revealed that TIM and CLSPN dissociate from stalled forks upon HU treatment in an ATR-dependent manner[49]. We also observed gradual dissociation of TIM and SDE2 at HU-stalled replication forks from iPOND, albeit with much slower kinetics than PCNA (Supplementary Fig 7e). Since these proteins are essential mediators of checkpoint activation, we presume that some dynamic changes are expected to occur within the replisome during fork remodeling, in order to accommodate new DNA replication intermediates required to protect damaged forks and to counteract fork collapse. ATR-mediated global fork slowdown and reversal that has been shown to occur upon DNA interstrand cross-link formation, may similarly contribute to HU-induced stalled and

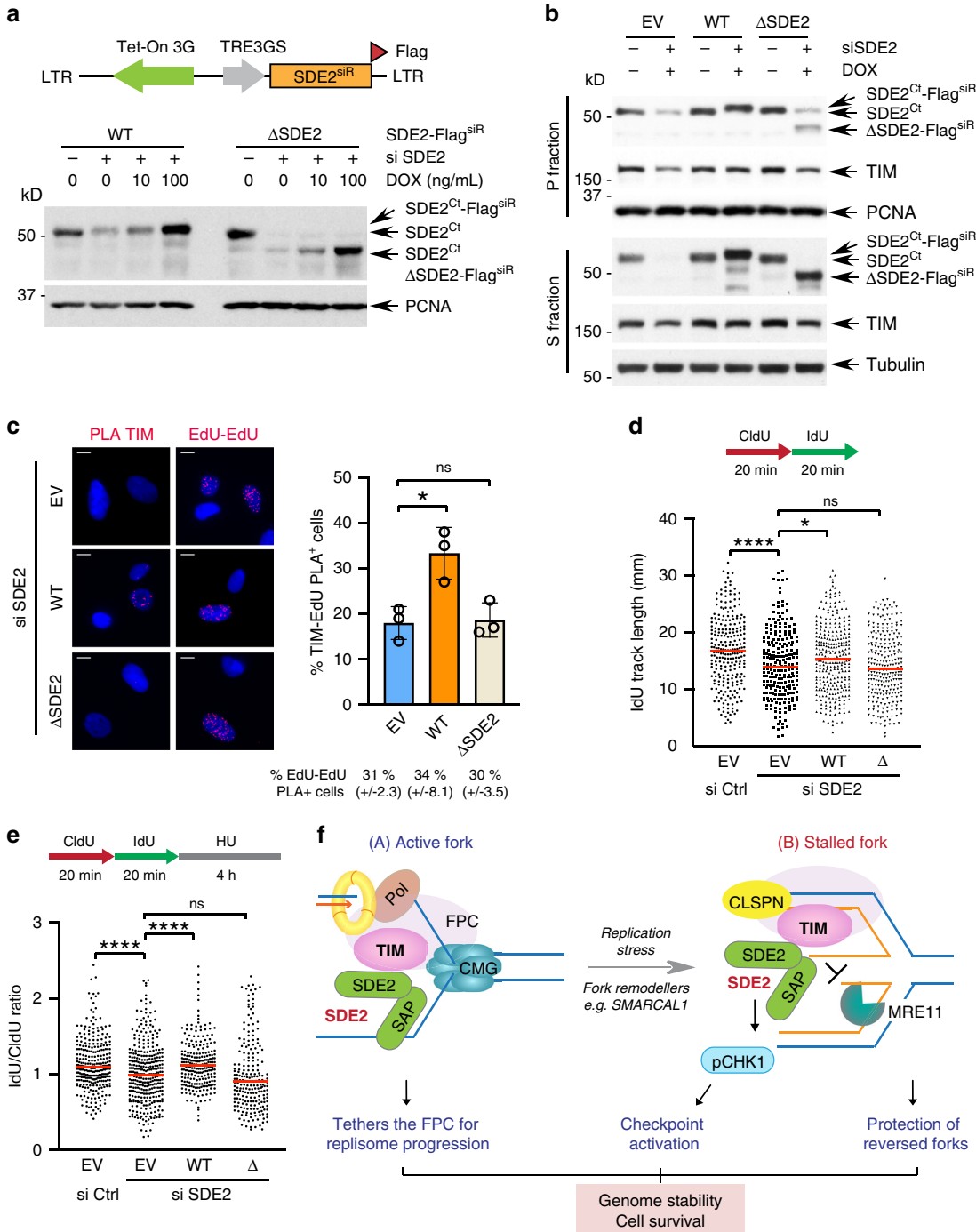

**Fig. 7 SDE2-TIM interaction is required for protecting replication forks. a** Top: schematic of the Retro-X Tet-One system to express siRNA-resistant (siR) SDE2-Flag WT or ΔSDE2 mutant under the control of the doxycycline (dox)-inducible $P_{TRE3GS}$ promoter. Bottom: induction of Flag-tagged SDE2 WT or ΔSDE2 in response to increasing dose of dox, following transfection of SDE2 siRNA (vs. control). **b** TIM levels in S and P fractions from U2OS cells reconstituted with SDE2-Flag WT or ΔSDE2. See Fig. S7A for quantification. **c** Left: representative images of TIM:EdU PLA foci in the Retro-X SDE2 WT or ΔSDE2 cells following SDE2 siRNA transfection and dox induction. Scale bar, 10 μm. Right: quantification of cells positive for TIM:EdU PLA foci. (>100 cells per condition, $n = 3$ biologically independent experiments, mean ± SD, *$P < 0.05$, Student's t-test). **d** Dot plot of the DNA fiber IdU track length from the Retro-X U2OS cells reconstituted with SDE2 WT or ΔSDE2. (>200 tracks per condition, $n = 2$ biologically independent experiments, ****$P < 0.0001$, *$P < 0.05$, Mann–Whitney). **e** Dot plot of the DNA fiber IdU/CldU track length ratio from the Retro-X U2OS cells treated with 4 mM HU for 4 h ($n = 3$ biologically independent experiments, ****$P < 0.0001$, Mann–Whitney). **f** A model depicting the role of SDE2 in promoting TIM-dependent fork protection, coordinated at both active and stalled forks. See discussion for details.

reversed forks[50]. Interestingly, elevation of reactive oxygen species caused by metabolic stress was shown to trigger dissociation of TIM from the replisome to modulate fork velocity, highlighting that multiple pathways exist and are required for the dynamic

change of the FPC to control fork integrity under replication stress[51]. Although the exact nature of the FPC role within a reversed fork remains elusive, it may act upon its alternative role of restricting nuclease activities to stabilize forks and suppress

extensive generation of ssDNA intermediates, which is distinct from its canonical role in coupling helicase-polymerase movement during unstressed conditions. We have previously shown that damage-dependent degradation of SDE2 is specifically required for the acute response to UVC damage[35]. Thus, although responses to HU and UVC damage may differ, it is possible that the modulation of TIM stability by SDE2 in a damage-dependent manner may further affect the FPC dynamics at stalled forks and engage it in previously unidentified function of fork protection.

An important question to pursue is whether TIM directly antagonizes MRE11 nuclease activity or indirectly contributes to fork protection by promoting the ATR checkpoint or stabilizing RAD51 filaments. ATR is known to regulate the activity of proteins involved in fork reversal and protection. Therefore, TIM in conjunction with CLSPN, may contribute to such processes by regulating the localization and activity of key players that process stalled forks[52–55]. Intriguingly, TIM is known to interact with PARP1, which has multiple roles in the processing of reversed forks through regulation of proteins such as XRCC1, RECQ1, and MRE11[40,41,56]. The PARP1-TIM complex may be required for the protection of reversed forks by modulating these protein activities, and SDE2 may contribute in the regulation of this process. We show that the C-terminus of TIM binds to both SDE2 and PARP1, so it is possible that their function may be cooperative.

**TIM as an alternative fork protection mechanism in cancer.** While fork reversal is considered an important step in the cellular response to replication stress, reversed forks can also become a subject of pathological degradation when not properly protected. To counteract detrimental degradation, BRCA2 stabilizes RAD51 nucleofilaments, thus preventing unrestricted nuclease activity from MRE11, CtIP, and EXO1[17,18,21,57]. Several independent mechanisms counteract nuclease activities in conjunction with BRCA2, such as suppression of DNA2 by ABRO1 and BOD1L[43,58]. Although we were unable to detect a statistically significant increase in resection of BRCA2 and TIM double knockdown cells compared to individually depleted cells, we do not exclude the possibility that shortening of ongoing replication tracks by TIM knockdown may preclude the recognition of additional shortening by the resection in DNA combing assays. For example, lack of synergy in fork degradation despite a loss of viability in double mutants was also reported in the deficiency of EXD2 nuclease in BRCA1/2 null backgrounds[59].

Therefore, we speculate that TIM-mediated fork protection may work together with BRCA2 to rescue stalled forks, both channeling to limit MRE11 nuclease activity, and a synthetic lethality may arise when the two distinct fork protection mechanisms are impaired simultaneously. Whether the FPC directly counteracts MRE11 activity is not known. Intriguingly, PARP1 and BRCA2 have been shown to cooperate to prevent MRE11-dependent degradation of stalled forks, and TIM may participate in this process by directly interacting with PARP1[60]. Tumors bearing BRCA1/2 mutations experience high levels of DNA replication stress, and therefore may exploit the TIM-mediated fork protection mechanism to salvage the damaged fork and alleviate replication stress. TIM is known to be overexpressed in diverse groups of cancer, indicating that TIM-dependent fork protection may confer a survival advantage to cancer cells with elevated replication stress. This oncogenic activity may negatively influence therapeutic responses to cytotoxic chemotherapies or PARP inhibitors especially in BRCA1/2-deficient conditions and allow cells to acquire chemoresistance by modulating relevant pathways.

In conclusion, our work elucidates a new mechanism in which replication fork integrity is preserved by the SDE2-TIM interaction and reveals the FPC as a new player in the protection of stalled forks from over-resection. Knowledge of how the FPC engages in replication fork reversal and fork stabilization mechanisms may reveal new therapeutic targets in order to induce synthetic lethality in cancer cells vulnerable to DNA replication stress.

## Methods

**Cell culture and cell lines.** U2OS and HEK293T cell lines were acquired from the American Tissue Culture Collection (ATCC). Cells were cultured in Dulbecco's Modified Eagle's Medium supplemented with 10% fetal bovine serum and 1% penicillin/streptomycin, following standard culture conditions and procedures.

**Generation of Retro-X Tet-On inducible cell lines overexpressing SDE2-Flag.** The retroviral plasmid pRetroX-TetOne puro was acquired from Clontech and amplified using NEB Stable competent E. coli (high efficiency). siRNA-1 resistant SDE2 WT or ΔSDE2 (internal deletion of amino acids 108–150) were subcloned into pRetroX-TetOne puro vector using EcoRI and BamHI restriction sites and PCR primers designed to add a C-terminal Flag tag. After pilot-testing the doxycycline-based induction of our construct using transient transfection, retroviruses were produced using the GP2-293 packaging cell line (Clontech), where pRetroX-TetOne puro empty vector (EV), SDE2 WT or ΔSDE2 constructs were co-transfected with the envelope vector pCMV-VSV-G, using Xfect™ transfection reagent (Clontech). U2OS cells were transduced for 16 h, using 8 μg/mL polybrene (Sigma-Aldrich). Puromycin selection (2 μg/mL) began 48 h post-infection and lasted for 3 days, until all non-transfected cells had died. After a week, cells had recovered and were tested to find the optimal doxycycline concentration to induce the transgene. In subsequent experiments, Retro-X cells were induced with 100 ng/mL of doxycycline diluted in culture media, for 48 to 72 h. For the indicated experiments, single-cell clones of Retro-X cells were used to improve leakiness by limited dilutions of the pools in 96-well plates.

**Generation of a stable cell line overexpressing SDE2-Flag-APEX2.** pcDNA3-APEX2-NES was a gift from Alice Ting (Addgene plasmid # 49386). The Flag-APEX2 tag was subcloned into the retroviral plasmid pMSCV-SDE2 using XhoI and EcoRI restriction sites. HEK293T cells were transfected with the retroviral plasmid pMSCV-SDE2-Flag-APEX2, along with the retroviral plasmids pCMV-Gag/Pol and pCMV-VSV-G. The viral supernatant was harvest 48 h post-transfection, and U2OS cells were infected for 16 h. Puromycin selection (2 μg/mL) began 24 h post-infection in growth medium and lasted for 2–3 days prior to further analysis.

**Plasmid constructions.** pcDNA3-SDE2-Flag and its mutants were previously described[34,35]. pcDNA4-Flag-TIMELESS-myc-6xHis and pcDNA3-Flag-TIPIN were a gift from Aziz Sancar (Addgene plasmids #22887 and #22889). pcDNA3.1-Flag-CLASPIN was a gift from Michele Pagano (Addgene plasmid #12659). Primers containing restriction sites were used to amplify cDNAs for subcloning and primers with mutations or deletions were used for site-directed mutagenesis (SDM). After amplification, the PCR products were purified using PCR purification (Qiagen), followed by restriction enzyme digestion, purification using gel extraction kit (Qiagen), and ligation. The ligated product was transformed into DH5α competent cells, and individual colonies were inoculated in Luria-Bertani (LB) media for DNA extraction using DNA miniprep or midiprep kits (Promega). All mutations were verified by Sanger DNA sequencing (Stony Brook University Genomic Facility). SDM primer information can be found in Table S1.

**DNA and siRNA transfection.** Unless otherwise stated, plasmid transfection was performed using GeneJuice (MilliporeSigma) according to the manufacturer's protocols. siRNA duplexes were transfected at 25 nM using Lipofectamine RNAi-MAX (Thermo Fisher). siRNA sequence information can be found in Table S1.

**Drug treatments.** Cycloheximide (CHX), dissolved at 100 mg/mL in DMSO, was used at 100 μg/mL in cell culture media for the indicated times. MG132 (Z-L-Leu-D-Leu-L-Leu-al) was dissolved at 10 mM in DMSO and cells were treated at 10 μM in cell culture media for the indicated times. Cells were synchronized at the G2/M boundary by treatment with 100 ng/mL nocodazole dissolved in DMSO. For UVC treatment, cells were first washed with PBS, and after suction, were irradiated at 40 J/m² UVC with the UV Stratalinker® 1800 (Stratagene), before replenishing them with fresh media and culturing them for the indicated times. HU was dissolved in water at 500 mM stock and diluted to either 2 mM for 4 h (pCHK1 experiments), 4 mM for 4 h (DNA fiber experiments), or 250 μM for 20 h (pRPA experiments) in cell culture media that was equilibrated overnight in the incubator. CPT was dissolved in DMSO and used at 100 nM in cell culture media for the indicated times. Chemicals and reagents Information can be found in Table S2.

**Western blotting.** Cells were collected by scraping or trypsinization, washed in ice-cold phosphate-buffered saline (PBS), and lysed in lysis buffer complemented with protease inhibitor cocktails (MilliporeSigma) and phosphatase inhibitor cocktails (Thermo Fisher) for 40 min on ice. NETN300 (300 mM NaCl, 50 mM Tris pH 7.5, 0.2 mM EDTA, 1% NP40) was used for whole-cell lysis and NETN150

(150 mM NaCl, 50 mM Tris pH 7.5, 0.2 mM EDTA, 1% NP40) was used for co-immunoprecipitation (IP) assays. The lysates were then cleared by centrifugation at 15,000 rpm at 4 °C for 10 min. The protein concentration of supernatants was measured using the Bradford assay so that 20–30 µg of protein was loaded onto an SDS-PAGE gel and transferred to PVDF membranes (MilliporeSigma). Membranes were incubated with the indicated primary and secondary antibodies, and HRP signal was detected by enhanced chemiluminescence (ECL) Western blotting substrates (Thermo Fisher) using Hyblot CL autoradiography film (Thomas Scientific) or the iBright imager (Thermo Fisher). Antibody information can be found in Table S3.

**Immunoprecipitation and subcellular fractions**. HEK293T cells transfected with the indicated plasmids were harvested 36–48 h post-transfection. Cells were lysed with NETN150 lysis buffer and incubated with pre-equilibrated anti-FLAG M2 affinity gels (Sigma-Aldrich), anti-HA magnetic beads (Thermo Fisher), or anti-c-Myc agarose beads (Sigma-Aldrich) for 4 h at 4 °C with gentle rocking. After incubation, the beads were washed 3–4 times in NETN150 lysis buffer, and protein complexes were eluted by boiling in 2× Laemmli sample buffer for 5 min. For subcellular fractionations, cells were collected by trypsinization, washed with ice-cold PBS, lysed in cytoskeleton (CSK) buffer (100 mM NaCl, 10 mM Tris-HCl pH 6.8, 300 mM sucrose, 3 mM MgCl$_2$, 1 mM EDTA, 1 mM EGTA, 0.1% Triton X-100) complemented with protease and phosphatase inhibitor cocktails and incubated on ice for 5 min. Lysates were pelleted at $1500 \times g$ for 5 min, and the supernatant, labeled as S, was collected and the concentration measured. Pellets, designated as P, were resuspended with a 1:1 ratio of PBS and 2× boiling buffer (50 mM Tris-HCl pH 6.8, 2% SDS, 850 mM β-mercaptoethanol) and boiled for 15 min.

**Recombinant protein purification and GST pull-down**. GST or GST-SDE2 was expressed using BL21 (DE3) cells by incubating at 37 °C to an OD$_{600}$ of 0.6, induced with 0.5 mM isopropyl β-D-1-thiogalactopyranoside (IPTG, Sigma-Aldrich), and then incubating at 30 °C for 6 h. After centrifugation, cells were resuspended in PBS with 1 mg/mL lysozyme and protease inhibitor, rocked at 4 °C for 40 min, and stored at −80 °C. After thawing, the cells were sonicated at 40% amplitude with 20 s on and 20 s off pulses for three cycles (Branson Digital Sonifier). Triton X-100 was added to 1% and the lysate was incubated at 4 °C for 1 h. Cell lysates were centrifuged at 14,000 rpm at 4 °C for 15 min, the supernatant filtered through a 0.22 µm PES filter (MilliporeSigma), and aliquots were stored at −80 °C. To purify, 1 mL of lysate was thawed on ice, diluted with 4 mL 0.5% Triton X-100 in PBS (PBS-T) and incubated with prepared glutathione sepharose beads (GE Healthcare) in a gravity column for 3 h at 4 °C. Columns were prepared by washing with 10 mL elution buffer (50 mM Tris-HCl pH 8.0, 150 mM NaCl, 0.1 mM EDTA, 10 mM reduced glutathione), 10 mL PBS, and 10 mL PBS-T. After incubation, the resin was allowed to settle, and the column was washed with 20 mL wash buffer (50 mM Tris-HCl pH 8.0, 150 mM NaCl, 0.1 mM EDTA) and the protein was eluted with elution buffer in 500 µL fractions. The fractions were visualized via Coomassie staining, and protein-containing fractions were pooled and dialyzed using 3.5 K MWCO SnakeSkin pleated dialysis tubing (Pierce) in 2 L dialysis buffer (50 mM Tris-HCl pH8.0, 1 mM DTT, 0.2 mM EDTA, 10% glycerol). Protein was recovered and stored at −80 °C until needed. For the pull-down, the purified recombinant protein was incubated with glutathione sepharose beads at 4 °C for 3 h. After washing with PBS, the beads were incubated with either 293 T cell lysates or in vitro translated products for 4 h at 4 °C followed by three washes. For in vitro translation, proteins were produced from pcDNA3-based plasmids at 30 °C for 90 min using the in vitro transcription & translation (IVTT) kit (Promega), following manufacturer's instructions.

**DNA fiber analysis**. Exponentially growing cells were pulse-labeled with 50 µM CldU for the indicated time, washed three times with PBS, then pulse-labeled with 250 µM IdU for the indicated time. In the case of resection studies, cells were further washed three times with PBS before replenishing media with 4 mM HU for 4 h. All necessary media was equilibrated overnight at 37 °C under 5% CO$_2$. Cells were harvested by trypsinization, 400,000 cells pelleted, and washed with PBS. DNA fibers were then prepared using the FiberPrep® DNA extraction kit and the FiberComb® Molecular Combing System (Genomic Vision, France), following manufacturer's instructions. In brief, the cells were washed again with PBS before being embedded in low-melting point agarose, and cast in a plug mold. After full solidification, plugs were digested overnight with proteinase K. Next day, the plugs were extensively washed prior to short melting and agarose digestion. The obtained DNA fibers were then combed onto silanized coverslips (Genomic Vision, France) that were subsequently baked for 2 h at 60 °C. DNA was denatured for 8 min using 0.5 M NaOH in 1 M NaCl. Subsequent immunostaining incubations were performed in humidified conditions at 37 °C. In short, coverslips were blocked with 1% BSA for 30 min, then two primary antibodies were diluted in 1% BSA (rat monoclonal anti-BrdU for CldU, 1:25, and mouse monoclonal anti-BrdU for IdU, 1:5) and incubated for 1 h. After washing the coverslips with PBS-Tween 0.05% (PBS-T), two secondary antibodies were diluted in 1% BSA (Alexa Fluor 594 goat anti-rat and Alexa Fluor 488 goat anti-mouse, 1:100) and incubated for 45 min. For some experiments, Alexa Fluor 488-conjugated goat anti-rat antibody and Alexa Fluor 568-conjugated goat anti-mouse antibody (1:100) were used to label CldU

and IdU, respectively (green followed by red). ssDNA counterstaining was performed with anti-ssDNA (MAB3034, Millipore Sigma) and Alexa Fluor 647 anti-mouse (1:100) antibodies. Coverslips were washed with PBS-T, dehydrated, and mounted onto microscopic glass slides using ProLong$^{TM}$ Gold Antifade overnight. DNA fibers were then imaged with the Eclipse Ts2R-FL inverted fluorescence microscope (Nikon) equipped with the Nikon DSQi2 digital camera, or LSM880 microscope (Carl Zeiss), and analyzed using Fiji and Prism (GraphPad).

**Proximity-ligation assay between proteins, and on nascent DNA**. To analyze the proximity of two proteins, the proximity-ligation assay (PLA) was performed, using the Duolink in situ red starter kit (Sigma-Aldrich) following the manufacturer's instructions. EdU labeling, click chemistry, and PLA were successively used to investigate the localization of proteins on nascent DNA, as previously described[15]. In brief, U2OS cells or U2OS Retro-X cells were transfected with the indicated siRNA oligos or DNA constructs, and the latter cell line was treated with or without 100 ng/mL doxycycline for 72 h. One day before EdU labeling, cells were seeded onto coverslips and cell culture medium was equilibrated overnight in a humidified incubator at 37 °C under 5% CO$_2$. EdU, dissolved in DMSO, was diluted into the pre-equilibrated media at 125 µM, and incubated with cells for 12 min. Following nascent DNA labeling, or in the case of PLA between two proteins, 36 h after transfection, cells were washed with cold PBS on ice and fixed with 4% paraformaldehyde for 10 min at RT. After three PBS washes, cells were kept in fresh PBS in a sealed plate at 4 °C protected from light. To recover the cells, coverslips were first washed with cold PBS, then permeabilized with 0.3% Triton X-100 in PBS, for 3 min on ice. For PCNA immunostaining, permeabilization/epitope exposure was instead performed with ice-cold 100% methanol for 10 min at −20 °C. Following permeabilization, cells were washed three times with PBS. For EdU labeling, cells were quickly blocked with 1% BSA in PBS for 10 min at RT. During this time, the Click-iT reaction cocktail was prepared for the conjugation of EdU alkyne with biotin-azide, following manufacturer's instructions (Thermo Fisher). In brief, the reaction cocktail was freshly prepared precisely in the following order of 1× Click-iT reaction buffer, 2 mM CuSO$_4$, 10 µM biotin-azide, and 1× Click-iT buffer additive. Coverslips were incubated with the reaction cocktail for 1 h at RT, in a humidified chamber protected from light. Following biotin-azide conjugation, coverslips were washed once with PBS, before proceeding to the PLA assay to investigate the proximity of SDE2 or TIM to biotin-conjugated nascent DNA. For PLA foci detection, coverslips were first blocked with a drop of blocking solution for 1 h. Subsequent steps were performed in a humidified chamber at 37 °C protected from light. Coverslips were then incubated with primary antibodies diluted in the antibody diluent as following: mouse anti-biotin (Jackson ImmunoResearch, #200-002-211, 1:2000), rabbit anti-biotin (Bethyl Laboratories, #A150-109A, 1:3,000), rabbit anti-SDE2 (Sigma, #HPA031255, 1:400), rabbit anti-TIMELESS (Bethyl Laboratories, #A300-961A, 1:500), mouse anti-MCM6 (Santa Cruz, #sc-393618, 1:500), mouse anti-Flag (Sigma, #F1804, 1:500), mouse anti-PCNA (Santa Cruz, #sc-56, 1:50), rabbit anti-GFP (Abcam, #ab290, 1:250). Next, the coverslips were washed twice with wash buffer A at RT then incubated with the PLUS and MINUS PLA probes for 1 h. After another two washes with wash buffer A at RT, coverslips were incubated with the ligation reaction for 30 min, before washed again twice with wash buffer A. The amplification reaction was then carried out on the coverslips for 100 min, before washing the coverslips twice with wash buffer B. After a final wash with 1:100 wash buffer B, coverslips were mounted in the wet in situ mounting medium with DAPI and fixed with nail polish. Coverslips were observed and imaged with the Eclipse Ts2R-FL inverted Nikon fluorescence microscope equipped with the Nikon DSQi2 digital camera. Fluorescence images were analyzed using NIS-Elements, Research BR software (Nikon), and quantification data were processed using Prism (GraphPad).

**iPOND**. iPOND was performed as previously described[61]. In brief, HEK293T cells were incubated with 10 µM 5-ethynyl-2′ deoxyuridine (EdU, Thermo Fisher) for 20 min. For pulse-chase experiments, EdU-labeled cells were washed with medium containing 10 µM thymidine and incubated with 10 µM thymidine (Sigma-Aldrich) for 0, 0.5, 2, and 4 h. To induce HU-stalled forks, cells were treated with 2 mM HU for the indicated time points following EdU labeling. After pulse-chase, cells were subsequently fixed in 1% formaldehyde for 20 min at RT, quenched using 0.125 M glycine, and washed three times with PBS. Cells were permeabilized with 0.25% Triton X-100 in PBS for 30 min at RT and pelleted. Permeabilization was halted with 0.5% BSA in PBS. Cells were pelleted again and washed with PBS. After centrifugation, cells were resuspended with a click reaction buffer (10 mM sodium-L-ascorbate, 20 µM biotin azide, and 2 mM CuSO$_4$) and incubated for 10 min at RT on a rotator. After centrifugation, the click reaction was halted using 0.5% BSA in PBS. Cells were pelleted and washed with PBS twice. Cells were resuspended in lysis buffer (50 mM Tris-HCl, pH 8.0, and 1% SDS) supplemented with protease inhibitors (aprotinin and leupeptin) and sonicated. Lysates were cleared and incubated with streptavidin-agarose beads (MilliporeSigma) at 4 °C overnight on a rotator. The beads were washed once each with lysis buffer and 1 M NaCl, and then twice with lysis buffer. To elute proteins bound to nascent DNA, 2X SDS Laemmli sample buffer was added to packed beads (1:1; v/v). Samples were incubated at 95 °C for 30 min, followed by Western blotting.

**Immunofluorescence**. Cells were seeded on coverslips at least 24 h before any drug treatment. To stop the treatment, cells were washed with cold PBS on ice, then fixed with 4% paraformaldehyde for 10 min at 4 °C. After three washes with PBS, cells were permeabilized with 0.3% Triton X-100 in PBS for 3 min on ice. Following permeabilization, cells were washed three times with PBS, and blocked for 45 min at RT using 5% BSA in PBS. Subsequently, cells were incubated with primary antibodies diluted in 1% BSA for 1 h at RT. After three PBS washes, cells were incubated with secondary antibodies coupled to fluorochromes diluted in 1% BSA for 45 min at RT. After three additional PBS washes, coverslips were mounted onto microscope slides using Vectashield mounting medium containing DAPI (Vector Lab). Coverslips were analyzed with an Eclipse Ts2R-FL inverted Nikon fluorescence microscope equipped with the Nikon DSQi2 digital camera. Fluorescence images were analyzed using Fiji, and quantification data were processed by Prism (GraphPad).

**Proximity biotinylation**. U2OS cells stably expressing SDE2-Flag-APEX2 were seeded one day before labeling. BP labeling was initiated by changing the medium to fresh medium containing 500 μM BP at 37 °C for 30 min. Hydrogen peroxide ($H_2O_2$) was then added at a final concentration of 1 mM with gentle agitation and incubated for 1 min at RT. The reaction was stopped by replacing medium with the quencher solution (5 mM Trolox, 10 mM sodium ascorbate, and 10 mM sodium azide in PBS). Cells were washed with the quencher solution three times before proceeding to imaging or Western blotting experiments. For Western blotting, cell pellets were lysed in NETN300 complemented with protease inhibitor and quencher solution. For streptavidin pull-down assay, streptavidin-agarose resin (Thermo Fisher) was washed once with PBS and then twice with NETN150, and incubated with cell lysates for 4 h at 4 °C on a rotator. The beads were subsequently washed four times with NETN150 lysis buffer, and biotinylated proteins were eluted by boiling the beads in 2× Laemmli sample buffer for 5 min. For the imaging of SDE2-Flag-APEX2, U2OS cells on coverslips were treated with BP and $H_2O_2$, and fluorescence labeling was performed by incubating coverslips with Alexa Fluor 594-coupled streptavidin (Thermo Fisher) in 1% BSA, incubated in a humidifying chamber at 37 °C for 45 min. Coverslips were analyzed with the Eclipse Ts2R-FL inverted Nikon fluorescence microscope equipped with the Nikon DSQi2 digital camera.

**BrdU staining for ssDNA detection**. Cells were grown on coverslips and incubated with 10 μM BrdU for 48 h before HU treatment at 2 mM for 4 h. Cells were permeabilized with 0.3% Triton X-100 in PBS for 3 min at 4 °C, washed three times in PBS, and fixed with 4% paraformaldehyde for 10 min at 4 °C. After three PBS washes, cells were blocked with 5% BSA, then incubated with mouse anti-BrdU (BU-1, 1:300) antibody in 1% BSA for 2 h. After three PBS washes, cells were incubated with 1:1000 Alexa Fluor 488 goat anti-mouse IgG for 45 min in 1% BSA. After three PBS washes, coverslips were mounted using Vectashield mounting medium containing DAPI (Vector Lab) and analyzed with the Eclipse Ts2R-FL inverted Nikon fluorescence microscope equipped with the Nikon DSQi2 digital camera. Corrected total cell fluorescence (CTCF) intensity was quantified and calculated using Fiji and analyzed with Prism (GraphPad).

**EdU staining and flow cytometry**. To label replicating cells, siRNA-transfected cells were incubated with 10 μM EdU (Thermo Fisher) for 30 min before harvest. Harvested cells were fixed with 4% paraformaldehyde for 15 min at RT, permeabilized by saponin-based permeabilization buffer (Thermo Fisher) for 15 min, and subjected to EdU-click reaction using Alexa Fluor 488 picolyl azide and click-iT Plus EdU flow cytometry assay kit (Thermo Fisher) following manufacturer's protocol. Cells were washed once and resuspended with 200 μg/mL PureLink$^{TM}$ RNase A and eBioscience$^{TM}$ 7-AAD viability staining solution (Thermo Fisher). After 30 min of incubation at 37 °C, cells were sorted with the Attune NxT acoustic focusing cytometer and analyzed with the Attune NxT software v2.7 (Thermo Fisher).

**Comet assay**. DNA comet assay was performed using the CometAssay kit (4250-050-K, Trevigen) according to the manufacturer's protocol. Twenty-five microliters of a cell suspension at $2 \times 10^5$ cells/mL were combined with 225 μL of low-melting agarose (1:10 ratio, v/v), and 50 μL were spread on comet slides (Trevigen). After solidification, the slides were immersed in cold lysing solution for 45 min at 4 °C and placed in freshly prepared alkaline unwinding solution (200 mM NaOH, 1 mM EDTA) for 20 min at RT. Electrophoresis of unwound DNA was performed at 21 V for 30 min. The slides were washed with dH$_2$O for 5 min, dehydrated with 70% ethanol for 5 min, dried, and stained with SYBR Gold (Thermo Fisher). Comet tails were examined using a Nikon Eclipse E600 fluorescence microscope and analyzed using Fiji to determine tail lengths. Up to 50 individual nuclei were evaluated per group.

**Cell survival assay**. Cells in 6-well plates were transfected with siRNA oligos, seeded on 96-well plates 48 h later, and cell viability was determined using the CellTiter-Glo luminescent cell viability assay (Promega) seven days post-transfection. Luminescence was measured using a GloMax Explorer microplate luminometer (Promega).

**Statistics and reproducibility**. Student's $t$-test was used to assess the statistical significance, using Prism (GraphPad). Unpaired $t$-tests were performed with a 95% confidence interval, using two-tailed $p$-values, unless stated otherwise. For the DNA combing assay, distribution of track length or ratio were tested using a two-side Mann–Whitney $U$-test, with a 95% confidence interval, using Prism (GraphPad), and where indicated, Student's $t$-test was used to compare medians of replicates. The exact $p$-values are listed in the Source Data file. For PLA assays, distributions of foci number per cell were also tested using the Mann–Whitney $U$-test. Western blotting, DNA combing, immunofluorescence, DNA comet assay, survival assay, and cell cycle analysis were representative of at least two, mostly three, biologically independent experiments, and showed reproducible results.

**Reporting summary**. Further information on research design is available in the Nature Research Reporting Summary linked to this article.

## Data availability
The data that support the findings of this study are available within the Article, Supplementary Information, or from the corresponding author upon reasonable request. The source data underlying Figs. 1c, e, h, 2a–h, 3a–g, 4a–f, 5a–g, 6a–h, 7a–e and Supplementary Figs. 1a–d, 2a–e, 3a–f, 4a–d, 4f, 5a–e, 6b–h, 7a–e are provided as a Source Data file. All data are available from the correspondent author upon reasonable request.

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

## Acknowledgements

We thank members in the Kim laboratory for helpful discussions and comments on the manuscript. This study was supported by the National Institutes of Health (R01CA218132 to H.K.), an American Cancer Society Research Scholar Grant (132235-RSG-18-037-DMC to H.K.), Carol M. Baldwin Breast Cancer Research Award (to H.K.), and the Korean Institute for Basic Science (IBS-R022-A1-2017 to O.D.S. and J.-E.Y.).

## Author contributions

J.R. and J.J.P. designed the project, performed and analyzed most of the experiments with the assistance of P.P.Z. and N.L. E.-A.L., J.Y., and J.-E.Y. performed DNA combing experiments (Figs. 4d–f, 6e, f) and analyzed data. S.H. performed iPOND experiments. A.S. W. performed some of the PLA assays. J.-E.Y. and O.S. supervised the work and assisted with experimental design and manuscript preparation. H.K. conceptualized and supervised the project, performed some of the experiments, and wrote the manuscript with the assistance of J.R. All authors read and approved the manuscript. These authors contributed equally: J.R. and J.J.P. These authors jointly supervised this work: J.-E.Y. and H.K.

## Competing interests

The authors declare no competing interests.
