## [Peer Review File · Nature Communications]

REVIEWER COMMENTS

Reviewer #1 (Remarks to the Author):

The stability of eukaryotic replication forks depends on the fork protection complex (FPC), which couples polymerases and helicases activities and plays a central role in the replication stress response. In human cells, the FPC is composed of Timeless (TIM), Tipin, Claspin and AND1. In this manuscript, Rageul and colleagues report the characterization of SDE2 as a novel regulator of the FPC that interacts physically with TIM and promotes its recruitment to the replisome. More specifically, the authors used different approaches (iPOND, EdU-PLA, and co-IPs with replisome components) to show that SDE2 localizes to nascent DNA. They also show that SDE2 regulates the stability of TIM in the nuclear fraction. As such, SDE2 depletion recapitulates most of the defects that are usually observed upon TIM deficiency. Importantly, the authors used TIM depletion as an internal control in most experiments, which is very useful to evaluate the quality of the methods and the data. Rageul et al. show that SDE2 knockdown leads to fork progression defects in normal conditions, increased fork resection (MRE11-dependent degradation of nascent DNA) and incomplete CHK1 activation upon replication stress induced by hydroxyurea (HU) or UV. The authors report that either MRE11 or SMARCAL1 depletion alleviates nascent strand degradation in the absence of SDE2, suggesting that SDE2 protects reversed forks from the nucleolytic activity of MRE11. SDE2 knockdown further impairs the ability of forks to restart after replication stress and consistently increased levels of ssDNA and DNA damages accumulate in SDE2-depleted cells exposed to HU or UV. Incidentally, this study also shows that TIM protects stalled forks from hyper-resection, which was not known. Overall, the manuscript is very clearly written and the data are of high quality. This function of SDE2 in the protection of stalled fork is novel and important. As such, this manuscript should be of general interest for those interested in DNA replication and genome maintenance. However, important issues need to be addressed prior to publication, most of which would not require additional experiments.

Comments:

1. The results shown in Fig. 1C (SDE2-EdU PLA) suggest that SDE2 is present at less than 5 foci per nuclei. Moreover, iPOND data (Fig. 1E) show that unlike TIM, part of SDE2 persists on nascent chromatin after the thymidine chase. In contrast, the impediment of replication fork progression in SDE2-depleted cells is rather global (all forks are slowing down; see Fig. 4A). Although this discrepancy could be explained by technical limitations of the PLA assay, can the authors formally rule out the possibility that SDE2 is present at only a subset of the forks and indirectly controls the function of FPC? Moreover, ref #48 shows by iPOND that SDE2 is present at untreated forks, but is no longer detected at HU-arrested forks. This is counterintuitive in light of the author's model that SDE2 and TIM protect stalled replication forks. These issues need to be discussed. Ideally, the authors could perform time-course experiments to follow the association of SDE2 to replication sites by PLA and iPOND.
2. Along the same line, the authors show that SDE2-depleted cells have more ssDNA due to hyper-resection, more DNA breaks (comet), increased phospho-RPA (S4-S8) and gamma-H2AX, but these cells are unable to activate CHK1 in HU. Again, this is counterintuitive, as the laboratory of Eric J. Brown has shown that the depletion of TIM-Tipin increases DNA damage markers (including pCHK1-S345) in unchallenged cells (Smith et al. J Cell Biol 2009). One likely explanation is that SDE2- or TIM-deficient cells have unstable forks, which leads to increased DNA damage and CHK1 activation by the DNA damage pathway. In HU, the decreased level of Claspin on chromatin would specifically reduce CHK1 activation at stalled forks (Fig. 5G), despite the accumulation of ssDNA and the increased ATR activation. Again, this possibility should be discussed in the manuscript.
3. Lopes and colleagues have recently shown that localized fork arrest caused by DNA lesions can induce a global decrease of fork speed (Mutreja et al. Cell reports 2018). Moreover, Debatisse and colleagues have shown that in CHK1-depleted cells, global fork slowdown results from DNA

damage signaling by ATM-CHK2 (Techer et al. Cell reports 2016). These local versus global effects on fork progression are difficult to separate and the authors should be more cautious about the interpretation of their data.

4. DNA combing experiments were performed in U2OS cells. Have these experiments been also performed in non-transformed cells, such as immortalized fibroblasts?

5. DNA combing is more accurate when DNA counterstaining is used. Depletion of TIM, or BRCA2 or FANCD2 were often used as internal controls, which is fine, but the authors would have made their case stronger by including DNA counterstaining. In particular, this would have allowed them to measure inter-origin distances to determine whether there is an inverse correlation between fork speed and origin firing, as reported by Blow, Debatisse and others. In addition, the authors could have estimated the rate of fork arrest/pausing by measuring sister forks asymmetry.

6. In figures 6E and 6F, DNA combing results are shown as pooled data from two independent experiments. This is not the best way to analyze the data. The authors should rather show the results of a representative experiment and show the others in supplemental data.

7. In figures 4D and 4E, the length of IdU tracks should be plotted as an indication of the speed of restarted forks, in addition to the frequency of fork restart after HU. This could be done by showing on the same graph the distribution of CldU track length (before HU) and the distribution of IdU tracks lengths (after HU).

8. In the WB of Figs 5C and 5E, the level of CHK1 is decreasing upon depletion of SDE2. Is it reproducible and significant?

9. The authors should cite and discuss a recent publication from the Branzei lab showing the role of AND1 in fork protection (Abe et al. Nat Com 2018). In this article And1 deficiency results in fork slowing. However, this slowdown does not depend on MRE11 activity. In addition they show by electron microscopy that in absence of AND1 there is an accumulation of ssDNA gaps at the level of replication forks. This ssDNA is exposed has a consequence of MRE11-mediated degradation of nascent DNA. Do the authors believe that fork slowing depends on MRE11 in SDE2-depleted cells?

10. TIM-Tipin depletion reduces the viability of breast cancer cells (Baldeyron et al. Mol Oncol 2015). Moreover, Claspin, TIM and CHEK1 are overexpressed in different cancers and contribute to cancer progression (Bianco et al. Nat Com 2019). Is it also the case for SDE2? How about SDE2 levels in cancer cells? Do they correlate with TIM levels?

11. In Fig. 2A, the IP of endogenous proteins is not convincing as the IgG samples do show signals for SDE2 and TIM. Authors should provide a more convincing blot.

12. In the quantification of Fig. 3B, the authors should include the levels of TIM in TIM-depleted cells in order to let the reader appreciate the background levels of TIM and the accuracy of the quantification.

13. The reference to Noguchi 2012 (p.14) is not properly formatted.

14. The view that SDE2 and TIM act in parallel pathways to BRCA2 to prevent fork resection is an interesting hypothesis but the authors do not provide direct evidence that the FPC counteract MRE11 activity, at least as BRCA2 and RAD51 do. Indeed, it could be that MRE11 is more active in SDE2- or TIM-depleted cells because forks are pausing more frequently, generating therefore more substrates for MRE11. The authors should therefore tone down their statement.

15. The Lukas lab has recently reported that TIM is displaced from the replisome in response to oxidative stress in order to reduce fork speed (Somyajit et al, Science 2017). The authors should

discuss these data in light of their new results.

16. The authors have previously reported that SDE2 is removed for UV-arrested forks by proteolytic degradation. How do they reconcile this mechanism with the fact that SDE2 is required to promote the FPC-dependent restart of stalled forks after HU?

Reviewer #3 (Remarks to the Author):

The manuscript from Rageul et al. describes the regulatory role of SDE2 in fork protection complex (FPC) component TIMELESS (TIM) protein's stability at chromatin. By using different endogenous and ectopic overexpressing systems, the authors showed SDE2 directly interacts with the C-terminal domain of TIM and regulate TIM's stability at the replication fork. There results section is mainly highlighting two functional observations, i) SDE2-TIM is important for fork progression and stalled fork recovery, ii) SDE2-TIM protects reversed fork by inhibiting MRE11 dependent excessive fork degradation. Notably, they bring up a new concept that FPC components are involved in the protection of reversed forks, and SDE2 regulates that dynamics.

Previous studies by this group highlighted SDE2 as a PCNA associated protein that regulates genomic integrity during replication stress, and its degradation leads to the replication stress response. Here, they extend the characterization of SDE2 to give more mechanistic detail and found SDE2 as an FPC regulator at damaged forks. In the manuscript, the regulatory role of SDE2 in TIM dynamics is restricted to stability per se. Although not comprehensively validated, probably the SDE2-TIM axis is vital in damaged reversed fork protection. Overall, the manuscript has a logical flow. Most experiments are well designed, and data are organized in a way to support the main conclusion. However, some issues need to be addressed.

Comments:

1. As authors showed, SDE2 depletion leads to slower replication, a phenotype known for TIM loss, EdU staining should go down as it affects DNA synthesis. Thus, EdU staining cannot be used in PLA to show reduced chromatin localization of FPC/TIM/MCM in the figures.
2. In figure 1B, one positive control of proteins such as PCNA, TIM with EdU, and one negative control with EdU will be more affirmative for PLA experiment. And EdU-EdU PLA in both control Vs. siSDE2 will show if there is a difference in DNA replication upon SDE2 knockdown.
3. Figure 3A fractionation experiment, in the presence of MG132, clearly shows that SDE2 regulates TIM's stability at chromatin to some extent, not localization. Again, in figure 5G increase of TIM upon HU treatment in SDE2 knockdown condition suggests SDE2 is not involved in TIM localization at damaged chromatin. The Author needs to emphasize that in the abstract rather than " SDE2 directly...TIMELESS (TIM) and enhances TIM stability and its localization to replication forks..". Also, the author should add the stability of the TIM Δ C1 construct in chromatin (P) fraction, which cannot bind to SDE2 to validate the observation.
4. To show, CMG complex and polymerase uncoupling pRPA can be used as an ssDNA marker rather than MCM-EdU PLA upon SDE2 knockdown.
5. Similarly, TIM Δ C1 O/E should have similar consequences in replication fork length and fork recovery upon HU treatment. As TIM Δ C1 can still bind to TIPIN but not SDE2, this will exquisitely prove the importance of SDE2-TIM interaction in fork dynamics.
6. Please add a scatter plot rather than a bar plot for the fork recovery assay in figure 4E, which gives a better representation of the population distribution.
7. In figure 5B, rather than total gamma-H2AX, S phase-specific damage should be measured.
8. Possibly, SDE2 mediated stabilization of TIM is happening at a reversed fork. To prove that, a rescue experiment with TIM Δ C1 and WT TIM in TIM KD condition may be useful in fork degradation assay, 6E/F.

9. In figure 7, is there any reason behind using Δ SDE2 (which is SDE2 domain deletion) rather than SDE2-KK (point mutation), as both cannot bind to TIM, according to figure 2D. In figure 7, the failure to rescue can also be attributed to an independent function of SDE2 and not through the SDE2-TIM axis.

10. Finally, in the model, there are some missing labels SDE2-SAP.

We would like to thank the reviewers for their expert evaluation of our manuscript entitled, “SDE2 integrates into the TIMELESS-TIPIN complex to protect stalled replication forks”. We have thoroughly addressed the comments raised, and as a result feel that our manuscript has been substantially improved. The following is a point-by-point response to the reviewers. The reviewers’ comments are in italic.

REVIEWER COMMENTS

Reviewer #1 (Remarks to the Author):

The stability of eukaryotic replication forks depends on the fork protection complex (FPC), which couples polymerases and helicases activities and plays a central role in the replication stress response. In human cells, the FPC is composed of Timeless (TIM), Tipin, Claspin and AND1. In this manuscript, Rageul and colleagues report the characterization of SDE2 as a novel regulator of the FPC that interacts physically with TIM and promotes its recruitment to the replisome. More specifically, the authors used different approaches (iPOND, EdU-PLA, and co-IPs with replisome components) to show that SDE2 localizes to nascent DNA. They also show that SDE2 regulates the stability of TIM in the nuclear fraction. As such, SDE2 depletion recapitulates most of the defects that are usually observed upon TIM deficiency. Importantly, the authors used TIM depletion as an internal control in most experiments, which is very useful to evaluate the quality of the methods and the data. Rageul et al. show that SDE2 knockdown leads to fork progression defects in normal conditions, increased fork resection (MRE11-dependent degradation of nascent DNA) and incomplete CHK1 activation upon replication stress induced by hydroxyurea (HU) or UV. The authors report that either MRE11 or SMARCAL1 depletion alleviates nascent strand degradation in the absence of SDE2, suggesting that SDE2 protects reversed forks from the nucleolytic activity of MRE11. SDE2 knockdown further impairs the ability of forks to restart after replication stress and consistently increased levels of ssDNA and DNA damages accumulate in SDE2-depleted cells exposed to HU or UV. Incidentally, this study also shows that TIM protects stalled forks from hyper-resection, which was not known. Overall, the manuscript is very clearly written and the data are of high quality. This function of SDE2 in the protection of stalled fork is novel and important. As such, this manuscript should be of general interest for those interested in DNA replication and genome maintenance. However, important issues need to be addressed prior to publication, most of which would not require additional experiments.

We appreciate the reviewer’s comment that our manuscript is clearly written, and contains data of high quality, and is of general interest to the field of DNA replication and genome stability. The issues raised by the reviewer are addressed below.

Comments:

1. The results shown in Fig. 1C (SDE2-EdU PLA) suggest that SDE2 is present at less than 5 foci per nuclei. Moreover, iPOND data (Fig. 1E) show that unlike TIM, part of SDE2 persists on nascent chromatin after the thymidine chase. In contrast, the impediment of replication fork progression in SDE2-depleted cells is rather global (all forks are slowing down; see Fig. 4A). Although this discrepancy could be explained by technical limitations of the PLA assay, can the authors formally rule out the possibility that SDE2 is present at only a subset of the forks and indirectly controls the function of FPC? Moreover, ref #48 shows by iPOND that SDE2 is present at untreated forks, but is no longer detected at HU-arrested forks. This is

counterintuitive in light of the author's model that SDE2 and TIM protect stalled replication forks. These issues need to be discussed. Ideally, the authors could perform time-course experiments to follow the association of SDE2 to replication sites by PLA and iPOND.

The conditions for the PLA assay at EdU-labeled replication forks that we established in our laboratory are determined by the specificity and sensitivity of antibodies. We have optimized conditions for each PLA assay in which no PLA foci are present in knocked-down cells. We generally observe fewer foci for the SDE2-EdU PLA in comparison to other proteins, and this is likely due to the adjustment of SDE2 primary antibody dilution to reduce background signals. Hence, we believe that the number of PLA foci does not reflect the functional property of a protein at replication forks but rather that it varies depending on the experimental conditions optimized for immunofluorescence.

While we cannot formally rule out the possibility that SDE2 may indirectly control the function of FPC based on the PLA assays, our data demonstrating a direct SDE2-TIM interaction and the impact of SDE2 loss on TIM stability and localization at replication forks suggest that SDE2 is a part of the FPC and that controls its integrity. While the iPOND results from Dungrawala et al., 2015 (now ref #49) showed that TIM and CLSPN (and SDE2) dissociate from EdU-labeled forks following HU damage, their disengagement is a slow process (mostly obvious after 8 h post HU), indicating that individual components in the FPC exhibit distinct patterns of remodeling at stalled forks. In the discussion (*track change, p16*), we now state that “*some dynamic changes are expected to occur within the replisome during fork remodeling, in order to accommodate new DNA replication intermediates required to protect damaged forks and to counteract fork collapse*”. As suggested, we also performed the time-course of iPOND at HU-induced stalled forks. Consistent with Dungrawala et al., 2015 and other iPOND data, PCNA is rapidly lost while FANCD2 is enriched at stalled replication forks. We did observe gradually decreased levels of TIM (and to a lesser extent, SDE2) within 6 h post HU treatment, indicating that the FPC may partially disengage from stalled forks and may adopt a new intermediate composition for the protection of stressed and reversed forks. We have included this new result in **Supplementary Fig. 7e**.

2. Along the same line, the authors show that SDE2-depleted cells have more ssDNA due to hyper-resection, more DNA breaks (comet), increased phospho-RPA (S4-S8) and gamma-H2AX, but these cells are unable to activate CHK1 in HU. Again, this is counterintuitive, as the laboratory of Eric J. Brown has shown that the depletion of TIM-Tipin increases DNA damage markers (including pCHK1-S345) in unchallenged cells (Smith et al. J Cell Biol 2009). One likely explanation is that SDE2- or TIM-deficient cells have unstable forks, which leads to increased DNA damage and CHK1 activation by the DNA damage pathway. In HU, the decreased level of Claspin on chromatin would specifically reduce CHK1 activation at stalled forks (Fig. 5G), despite the accumulation of ssDNA and the increased ATR activation. Again, this possibility should be discussed in the manuscript.

Smith et al. (2009) showed that deficiency of TIM causes ssDNA accumulation and an increased dependence on ATR for continuous DNA synthesis in unchallenged conditions. We agree with the proposed model that the TIM-TIPIN complex plays a dual role at DNA replication forks, by preventing accumulation of ssDNA as a replisome component, which otherwise would activate ATR to sustain continuous DNA synthesis (i.e. functioning upstream of ATR) and working as a localized adapter to transmit the ATR signal to promote CHK1 phosphorylation at stalled forks during DNA replication stress (i.e. functioning downstream of ATR). We have included this point in Figure 5 (*track change p. 11*) and cited the paper (**Ref #42**).

3. Lopes and colleagues have recently shown that localized fork arrest caused by DNA lesions can induce a global decrease of fork speed (Mutreja et al. Cell reports 2018). Moreover, Debatisse and colleagues have shown that in CHK1-depleted cells, global fork slowdown results from DNA damage signaling by ATM-CHK2 (Techer et al. Cell reports 2016). These local versus global effects on fork progression are difficult to separate and the authors should be more cautious about the interpretation of their data.

We agree with the reviewer that impaired fork progression in SDE2- or TIM-depleted cells may be a combination of local and global effects on disrupting DNA replication activity, along with deregulated replication checkpoint activity mediated by ATR and/or ATM. As discussed in the point #2, ATR signaling may be activated to exert global effects on fork slowdown and reversal in response to the replication stress caused by TIM deficiency. To help interpret our data on the regulation of fork protection by TIM, we have included a sentence in the discussion, alerting readers to the possibility that the ATR-mediated global fork slowdown and reversal, characterized in DNA interstrand cross-links (Mutreja et al., 2018), may similarly contribute to HU-induced stalled and reversed forks (**Ref #50, track change p.16**).

4. DNA combing experiments were performed in U2OS cells. Have these experiments been also performed in non-transformed cells, such as immortalized fibroblasts?

We additionally performed DNA combing experiments in non-transformed BJ-TERT cells and obtained similar results; knocking down SDE2 or TIM significantly reduced replication track lengths, indicating that SDE2 and TIM are required for efficient fork progression in both non-cancerous and cancer cell lines. This result is included as new **Supplementary Fig. 4a**.

5. DNA combing is more accurate when DNA counterstaining is used. Depletion of TIM, or BRCA2 or FANCD2 were often used as internal controls, which is fine, but the authors would have made their case stronger by including DNA counterstaining. In particular, this would have allowed them to measure inter-origin distances to determine whether there is an inverse correlation between fork speed and origin firing, as reported by Blow, Debatisse and others. In addition, the authors could have estimated the rate of fork arrest/pausing by measuring sister forks asymmetry.

As suggested, we used ssDNA counterstaining together with DNA combing to identify adjacent origins and measured inter-origin distance in U2OS cells. In this scheme, CldU (green) is followed by IdU (red) labeling, together with DNA staining with an anti-ssDNA antibody. This analysis showed that a lack of SDE2 or TIM resulted in shorter inter-origin distance, indicative of an increase in new origin firing and fork slow-down under stress conditions (New **Fig. 4d**). We also analyzed fork asymmetry in SDE2 or TIM knocked-down U2OS cells by dual-labeling replication tracks (CldU: red -> IdU: green) and identifying sister forks from initiation events (i.e. green-red-green tracks only). The forks were scored as asymmetric if the ratio between the left fork length and the right fork length deviated by more than 25%. The analysis demonstrated that knocking down SDE2 or TIM markedly increases fork asymmetry of bidirectional forks, suggesting that replication forks within the same genomic context fail to progress normally and experience frequent stalling. We obtained similar results in BJ-TERT cells. Together, these data further support the role of SDE2 and TIM in promoting efficient fork progression. The fork asymmetry results acquired from U2OS cells are included in new **Fig. 4c**.

6. In figures 6E and 6F, DNA combing results are shown as pooled data from two independent experiments. This is not the best way to analyze the data. The authors should rather show the results of a representative experiment and show the others in supplemental data.

We now show representative results in the main figure panel, replacing pooled data (**New Fig. 6e & 6f**). We are not showing the other data set in supplemental figures due to space limitation, but we have acquired similar results.

7. In figures 4D and 4E, the length of IdU tracks should be plotted as an indication of the speed of restarted forks, in addition to the frequency of fork restart after HU. This could be done by showing on the same graph the distribution of CldU track length (before HU) and the distribution of IdU tracks lengths (after HU).

As suggested, DNA combing data were reanalyzed, and the distribution of CldU and IdU track lengths was depicted as a measure of fork recovery speed in order to determine any delay of fork restart and subsequent progression. The results show that knockdown of SDE2, TIM, individually or together, leads to a significant decrease of IdU track lengths, further supporting the notion that fork recovery is compromised in the absence of SDE2 and TIM. The bar graph of percentage of replication fork recovery is moved to **Fig. 4f** (used to be Fig. 4e), while a new result of track length distribution is included in **Supplementary Fig. 4e**, along with the immunoblots that used to be in the main panel (now **Supplementary Fig. 4f**).

8. In the WB of Figs 5C and 5E, the level of CHK1 is decreasing upon depletion of SDE2. Is it reproducible and significant?

We do sometimes observe changes of total CHK1 levels upon knockdown of SDE2. However, quantification from independent experiments indicated that there is no statistical significance to this observation (Quantification is shown in **Figure R1**). It is possible that increased stress due to the disruption of normal DNA replication may cause some degree of changes in cellular CHK1 levels transcriptionally or post-translationally, but our data do not provide any conclusive evidence in that regard.

9. The authors should cite and discuss a recent publication from the Brnzei lab showing the role of AND1 in fork protection (Abe et al. Nat Com 2018). In this article And1 deficiency results in fork slowing. However, this slowdown does not depend on MRE11 activity. In addition they show by electron microscopy that in absence of AND1 there is an accumulation of ssDNA gaps at the level of replication forks. This ssDNA is exposed has a consequence of MRE11-mediated degradation of nascent DNA. Do the authors believe that fork slowing depends on MRE11 in SDE2-depleted cells?

Thank you for sharing your insight into the SDE2 phenotype in DNA replication. As suggested, we have cited Abe et al., 2018 in the discussion and included a sentence to highlight the scaffolding role of the FPC at replication forks to maintain fork speed and prevent fork resection (**Ref. #47**). As an ancillary protein of the TIM-TIPIN complex, it is plausible that AND-1 may share many roles of the FPC in the replisome, thus exhibiting similar phenotypes as SDE2 and TIM do. However, we noticed several differences in how SDE2 and And-1 affect fork progression. First, overall EdU incorporation rate was not affected in And-1 depletion in chicken DT40 cells, indicating that bulk DNA synthesis is not affected. By contrast, our result shows that SDE2 and TIM knockdown impairs EdU incorporation (Supplementary Fig. 4b, c). Second, while Tipin and And-1 foci colocalize with EdU foci in DT40 cells, And-1 depletion did not abrogate the localization of Tipin and Claspin to replication forks. Third, cells reconstituted with the HMG domain-deleted And-1 mutant exhibited less profound reduction in replication fork speed compared to the Tipin and Claspin mutants. Therefore, despite And-1 being a critical part of the replisome, phenotypes of And-1 depletion may be more relevant to its function in bridging Pol α with the CMG helicase complex, thus promoting priming of DNA replication initiation and Okazaki fragment synthesis. The observation that the formation of long ssDNA gaps proximal to the replication fork junction upon And-1 depletion is suppressed by the inhibition of the MRE11 nuclease activity may be more directly related to its regulation of Pol α . Therefore, while we do not exclude the possibility that slow fork progression in SDE2-depleted cells may result from deregulated MRE11 activity toward replication forks, we believe that the phenotype, which is shared with TIM deficiency, results from a more direct consequence of uncoupling DNA polymerases and helicases due to the loss of the TIM-TIPIN heterodimeric complex that tethers the replisome components.

10. TIM-Tipin depletion reduces the viability of breast cancer cells (Baldeyron et al. Mol Oncol 2015). Moreover, Claspin, TIM and CHEK1 are overexpressed in different cancers and contribute to cancer progression (Bianco et al. Nat Com 2019). Is it also the case for SDE2? How about SDE2 levels in cancer cells? Do they correlate with TIM levels?

As the reviewer noted, the role of TIM in tumorigenesis, including bypass of senescence for oncogenic transformation and sustained tumor cell survival, is now being appreciated (Bianco et al., 2019). We are also interested in this aspect of SDE2 function and are actively engaged in studying SDE2 levels in tumors. Our preliminary data demonstrate that SDE2 is upregulated in breast cancer (from a publicly available TCGA dataset) and that this upregulation is more pronounced in basal-like breast cancers (**Figure R2**). The majority of tumors expressing basal markers are triple-negative breast cancer, with similar molecular features to the tumors arising in the carriers of BRCA1/2 mutations, suggesting that TIM and its regulator SDE2 may compensate for the BRCA1/2 deficiency in replication fork stabilization or related DNA repair processes. Nevertheless, we believe this aspect of SDE2/TIM function is beyond the scope of this current study, and will report this observation in a separate future manuscript.

11. In Fig. 2A, the IP of endogenous proteins is not convincing as the IgG samples do show signals for SDE2 and TIM. Authors should provide a more convincing blot.

We have repeated the experiment and now include an immunoblot of better quality in Fig. 2a.

12. In the quantification of Fig. 3B, the authors should include the levels of TIM in TIM-depleted cells in order to let the reader appreciate the background levels of TIM and the accuracy of the quantification.

We have included quantification of TIM in TIM-depleted cells as a control in **Fig. 3b**.

13. The reference to Noguchi 2012 (p.14) is not properly formatted.

The citation has been updated with proper formatting (**Ref. #25**).

14. The view that SDE2 and TIM act in parallel pathways to BRCA2 to prevent fork resection is an interesting hypothesis but the authors do not provide direct evidence that the FPC counteract MRE11 activity, at least as BRCA2 and RAD51 do. Indeed, it could be that MRE11 is more active in SDE2- or TIM-depleted cells because forks are pausing more frequently, generating therefore more substrates for MRE11. The authors should therefore tone down their statement.

We have edited our statement in the discussion to accommodate the reviewer's comment (track change, p18).

15. The Lukas lab has recently reported that TIM is displaced from the replisome in response to oxidative stress in order to reduce fork speed (Somyajit et al, Science 2017). The authors

should discuss these data in light of their new results.

We have cited the Somyajit et al, (2017) and highlighted in the discussion that multiple pathways exist to modulate the dynamic change of the FPC to control fork integrity during replication stress. In the context of our proposed model, dynamic changes are expected to occur within the replisome during fork remodeling, in order to accommodate new DNA replication intermediates required to protect damaged forks and counteract fork collapse (**Ref. #51, track change, p17**).

16. The authors have previously reported that SDE2 is removed for UV-arrested forks by proteolytic degradation. How do they reconcile this mechanism with the fact that SDE2 is required to promote the FPC-dependent restart of stalled forks after HU?

We have previously shown that SDE2 is degraded specifically in the chromatin-enriched fraction in response to UVC damage (Rageul et al, 2019). Intriguingly, we did not observe noticeable chromatin-associated degradation in response to other types of damage, including HU, indicating that specific DNA damage signaling may govern the degradation of SDE2. We have not identified what triggers SDE2 degradation specifically in response to UVC damage, but one interpretation would be that UVC lesions may physically hinder the progression of a DNA polymerase while HU slows down its progression due to the shortage of nucleotide pools. UVC lesions are often addressed by translesion DNA synthesis (TLS), which involves polymerase switching of replicative polymerases to TLS polymerases, and acute SDE2 degradation within the replisome may trigger (or contribute to) drastic change of the replisome necessary for engaging TLS. This mode of change in the replisome may be a different response at HU-stalled forks. Hence, while SDE2 may regulate the dynamic remodeling of the FPC via TIM regulation at HU-stalled forks, the mechanism could be distinct and may not involve proteolytic degradation to remove SDE2 from DNA. We clarified in the discussion that the role of SDE2 at stalled forks may differ depending on the source of genotoxic stresses (*track change, p17*).

Reviewer #3 (Remarks to the Author):

The manuscript from Rageul et al. describes the regulatory role of SDE2 in fork protection complex (FPC) component TIMELESS (TIM) protein's stability at chromatin. By using different endogenous and ectopic overexpressing systems, the authors showed SDE2 directly interacts with the C-terminal domain of TIM and regulate TIM's stability at the replication fork. There results section is mainly highlighting two functional observations, i) SDE2-TIM is important for fork progression and stalled fork recovery, ii) SDE2-TIM protects reversed fork by inhibiting MRE11 dependent excessive fork degradation. Notably, they bring up a new concept that FPC components are involved in the protection of reversed forks, and SDE2 regulates that dynamics.

Previous studies by this group highlighted SDE2 as a PCNA associated protein that regulates genomic integrity during replication stress, and its degradation leads to the replication stress response. Here, they extend the characterization of SDE2 to give more mechanistic detail and found SDE2 as an FPC regulator at damaged forks. In the manuscript, the regulatory role of SDE2 in TIM dynamics is restricted to stability per se. Although not comprehensively validated, probably the SDE2-TIM axis is vital in damaged reversed fork protection. Overall, the manuscript has a logical flow. Most experiments are well designed, and data are organized in a way to support the main conclusion. However, some issues need to be addressed.

We are pleased to know that our study brings up a new concept of the FPC in the protection of reversed forks and that our manuscript is well designed and organized.

Comments:

1. As authors showed, SDE2 depletion leads to slower replication, a phenotype known for TIM loss, EdU staining should go down as it affects DNA synthesis. Thus, EdU staining cannot be used in PLA to show reduced chromatin localization of FPC/TIM/MCM in the figures.

The reviewer made a valid point. Indeed, we observe less EdU staining in cells knocked down for SDE2 or TIM by two different methods: first, by EdU-EdU PLA (e.g. Figs. 3f, g), and second, by flow cytometry (e.g. Supplementary Figures 4b, c). In order to see the true effect of SDE2 or TIM knock-down on either TIM or MCM6 presence at nascent DNA with EdU labeling, and to account for an indirect effect caused by the decrease in EdU incorporation, we normalized the data by dividing the percentage of positive cells by the percentage of cells in S-phase (i.e. EdU-EdU PLA positive cells). This method of analysis has been previously described in Roy et al., 2018. We further clarified this normalization method in the legends of the corresponding figures. Additionally, we do not think that our results are an indirect effect of the decrease in EdU incorporation because the concentration of EdU used is high (125 μ M, compared to the usual 10 μ M concentration), and the treatment is brief (8-12 min). These conditions are based on Roy et al., 2018, and allow for an optimal EdU incorporation frequency for reliable PLA signals.

2. In figure 1B, one positive control of proteins such as PCNA, TIM with EdU, and one negative control with EdU will be more affirmative for PLA experiment. And EdU-EdU PLA in both control Vs. siSDE2 will show if there is a difference in DNA replication upon SDE2 knockdown.

As the reviewer suggested, we have included a PCNA-EdU PLA experiment as a positive control for the SIRF assay. We have also performed PLA assays either without biotin antibody (PCNA only) or without EdU incubation as negative controls to confirm the specificity of the PLA signals (New **Supplementary Fig. 1a**). In addition, as explained above, we routinely analyze data by normalizing the percentage of PLA positive cells based on EdU-EdU PLA positive cells from the same experiments and confirm that our positive results are not due to the disruption of DNA replication. For clarification, we have included the percentage of EdU-EdU PLA positive cells in **Fig. 1c** and also present the normalized data below for reference (**Figure R3**).

3. Figure 3A fractionation experiment, in the presence of MG132, clearly shows that SDE2 regulates TIM's stability at chromatin to some extent, not localization. Again, in figure 5G increase of TIM upon HU treatment in SDE2 knockdown condition suggests SDE2 is not involved in TIM localization at damaged chromatin. The Author needs to emphasize that in the abstract rather than "SDE2 directly....TIMELESS (TIM) and enhances TIM stability and its localization to replication forks..". Also, the author should add the stability of the TIM Δ C1 construct in chromatin (P) fraction, which cannot bind to SDE2 to validate the observation.

We have modified the abstract to reflect the reviewer's suggestion and avoid any confusion.

"SDE2 directly interacts with the FPC component TIMELESS (TIM) and enhances its stability, thereby aiding TIM localization at replication forks and the coordination of replisome progression."

Additionally, we showed that Flag-TIM Δ C1 mutant degrades more rapidly than wild-type upon treatment with cycloheximide, further suggesting that the interaction between SDE2 and TIM is required for stabilizing TIM. This new result is included in **Supplementary Fig. 3b**.

4. To show, CMG complex and polymerase uncoupling pRPA can be used as an ssDNA marker rather than MCM-EdU PLA upon SDE2 knockdown.

Using a number of approaches, including Western blotting, flow cytometry, immunofluorescence, and iPOND, we have shown that SDE2 depletion leads to increased pRPA levels (Figs. 6a, 6b, 6c, 6d). We further showed the accumulation of ssDNA revealed by native BrdU staining (Supplementary Fig. 5c), indicating that increased ssDNA formation is associated with upregulated pRPA levels. We have specifically devised an MCM6-EdU PLA assay to better visualize an uncoupling event of the helicase (i.e. MCM6) and polymerase (i.e. EdU incorporation to newly synthesized DNA) activities. By contrast, pRPA levels may not only indicate accumulation of ssDNA but also reflect general activation of DNA replication stress response signaling.

To further confirm the reliability of our assay, we have added an additional control PLA experiment, in which MCM6-EdU PLA foci positive cells were scored before and after HU treatment. The result showed a significant reduction of PLA positive cells after HU-induced fork stalling, supporting the notion that decreased PLA signals represent a helicase-polymerase uncoupling and ssDNA accumulation (new **Supplementary Fig. 3f**).

5. Similarly, TIM Δ C1 O/E should have similar consequences in replication fork length and fork recovery upon HU treatment. As TIM Δ C1 can still bind to TIPIN but not SDE2, this will exquisitely prove the importance of SDE2-TIM interaction in fork dynamics.

We thought that reconstitution of the TIM Δ C1 mutant into a TIM-deficient background rather than exogenous overexpression is a better way to evaluate the role of the SDE2-TIM interaction for preserving fork integrity. The structure-function analysis was performed in response to the subsequent point 8, and the new result is included in **Supplementary Fig. 7d**. We show that the TIM Δ C1 mutant fails to rescue the excessive fork resection of TIM-depleted cells, suggesting that the SDE2-TIM interaction is required for the protection of HU-induced stalled forks.

6. Please add a scatter plot rather than a bar plot for the fork recovery assay in figure 4E, which gives a better representation of the population distribution.

A similar point was raised by the reviewer #1. Please refer to our response to the point #1-7. A new result of track length scatter plot is included in **Supplementary Fig. 4e**.

7. In figure 5B, rather than total gamma-H2AX, S phase-specific damage should be measured.

As suggested, we evaluated S phase-specific γ H2AX foci by marking replicating cells with EdU that is fluorescently labeled by a click reaction. Our immunofluorescence analysis showed that γ H2AX foci in TIM-depleted cells are specifically enriched in EdU-positive cells, and not their EdU-negative counterparts, confirming an increase in replication-associated DNA damage in the absence of the FPC. This new result is included in **Supplementary Fig. 5b**.

8. Possibly, SDE2 mediated stabilization of TIM is happening at a reversed fork. To prove that, a rescue experiment with TIM Δ C1 and WT TIM in TIM KD condition may be useful in fork degradation assay, 6E/F.

As suggested, we performed a fork degradation assay using cells depleted of endogenous TIM by siRNA and complemented with siRNA-resistant Flag-tagged TIM WT or Δ C1 mutant that cannot interact with SDE2. Our result showed that the TIM Δ C1 mutant fails to rescue the fork resection defect in TIM-depleted cells, further supporting our notion that the SDE2-TIM interaction is required for the protection of damaged forks. This new result is included in **Supplementary Fig. 7d**.

9. In figure 7, is there any reason behind using Δ SDE2 (which is SDE2 domain deletion) rather than SDE2-KK (point mutation), as both cannot bind to TIM, according to figure 2D. In figure 7, the failure to rescue can also be attributed to an independent function of SDE2 and not through the SDE2-TIM axis.

We initially created the Δ SDE2 mutant to test the hypothesis that the conserved SDE2 domain is important for TIM interaction and thus regulation of its function. Generation of Retro-X Tet-On inducible SDE2 cell lines takes considerable amount of time and effort, and we had developed the SDE2 complementation system at an early stage using the Δ SDE2 mutant before we discovered the conserved lysine residues. Since SDE2 undergoes endolytic cleavage to become a short form, SDE2^{Ct}, subsequent SDE2 domain deletion accounts for a relatively small truncation of the SDE2 polypeptide, and we believe that the phenotype is specific, not due to aberrant structure of the mutant. Nevertheless, we cannot entirely rule out the possibility that the phenotypes might be attributed to an independent function of the SDE2 domain. To substantiate the idea that the KK (K132A/K135A) mutant behaves similarly to Δ SDE2, we also performed the experiment to test whether the KK mutation is sufficient to abrogate the ability of SDE2 to promote TIM stability. Indeed, similarly to what we observed in Fig. 3d with the Δ SDE2 mutant, the SDE2 KK mutant failed to increase cellular levels of Flag-TIM, suggesting that the KK mutation disrupts the SDE2-TIM interaction and behaves similarly to the Δ SDE2 mutant. This result, together with other data in Figs. 3 and 7 using the Δ SDE2 mutant, further supports the importance of the SDE2-TIM axis as a means to sustain the integrity of TIM and its function at stalled forks. The new KK result is included in **Supplementary Fig. 3c**.

10. *Finally, in the model, there are some missing labels SDE2-SAP.*

The missing labels were corrected.

REVIEWERS' COMMENTS

Reviewer #1 (Remarks to the Author):

The authors have done a great job revising this manuscript. The new data they provide significantly strengthened their message. In my opinion, the manuscript is now suitable for publication.

Reviewer #3 (Remarks to the Author):

The authors have addressed my concerns!

REVIEWERS' COMMENTS

Reviewer #1 (Remarks to the Author):

The authors have done a great job revising this manuscript. The new data they provide significantly strengthened their message. In my opinion, the manuscript is now suitable for publication.

Reviewer #3 (Remarks to the Author):

The authors have addressed my concerns!

Thank you for your constructive criticism and suggestions to help strengthen our manuscript.